# Dynamic regulation of mRNA acetylation at synapses by spatial memory in mouse hippocampus

**Hai-Qian Zhou[1,2†], Zhen Zhu[1†], Jia-Wei Zhang[1†], Wei-Peng Lin[1,3,4], Hao-JY Jin[1], Yang-Yang Ding[1], Shuai Liu[1,3*], Dong-Min Yin[1,3,5*]**

[1]Key Laboratory of Brain Functional Genomics, Ministry of Education and Shanghai, Affiliated Mental Health Center, School of Life Science, East China Normal University, Shanghai, China; [2]Department of Neurology, Shanghai Fifth People's Hospital, Fudan University, Shanghai, China; [3]Shanghai Changning Mental Health Center, Affiliated Mental Health Center of East China Normal University, Shanghai, China; [4]Chongqing Key Laboratory of Precision Optics, Chongqing Institute of East China Normal University, Chongqing, China; [5]NYU-ECNU Institute of Brain and Cognitive Science at NYU Shanghai, Shanghai, China

**\*For correspondence:**
sliu@psy.ecnu.edu.cn (SL);
dmyin@brain.ecnu.edu.cn (D-MY)

[†]These authors contributed equally to this work

**Competing interest:** The authors declare that no competing interests exist.

## eLife Assessment

Recent studies have shown that mRNA can be acetylated (ac4c), altering mRNA stability and translation efficiency; however, the role of mRNA acetylation in the brain remains unexplored. In this **important** study, the authors demonstrate that ac4c occurs in synaptically localised mRNAs, mediated by NAT10. Conditional reduction of NAT10 protein levels led to decreases in ac4c of mRNAs and deficits in synaptic plasticity and memory. These **solid** results suggest that mRNA acetylation may play a role in memory consolidation.

**Abstract** Precise regulation of protein synthesis is critical for brain functions such as long-term memory, and its dysregulation is implicated in numerous memory disorders. While mRNA methylation, such as *N6*-methyladenosine (m6A), has been widely studied in memory, the role of mRNA acetylation remains largely unknown. *N4*-acetylcytidine (ac4C), the only known form of RNA acetylation in eukaryotes, promotes mRNA stability and translation. In this study, we identified the ac4C epitranscriptome in mouse hippocampal homogenates and synaptosomes through ac4C-RNA immunoprecipitation followed by next-generation sequencing (acRIP-seq). The Morris water maze was employed to induce and evaluate memory acquisition and forgetting processes. We show that synaptic ac4C levels are dynamically regulated, increasing after memory formation and returning to baseline after natural forgetting. The dynamic changes of ac4C-mRNAs regulated by memory were validated by ac4C dot-blot, liquid chromatography–tandem mass spectrometry, and acRIP-qPCR analysis. We further demonstrate that *N*-acetyltransferase 10, the ac4C writer, in mouse hippocampus is important for spatial memory via regulating memory-related mRNAs, proteins, and ultimately synaptic plasticity. Lastly, we generated a freely accessible website (http://ac4catlas.com) that included the dataset of ac4C epitranscriptome in mouse hippocampus. Altogether, these results demonstrate that dynamic and localized mRNA acetylation is important for synaptic plasticity and memory.

## Introduction

Spatiotemporal regulation of protein synthesis is essential for brain functions, particularly learning and memory (*Shrestha and Klann, 2022*; *Besse and Ephrussi, 2008*). For instance, local protein synthesis near stimulated synapses plays critical roles in synaptic plasticity and memory consolidation (*Martin et al., 2000*; *Sutton and Schuman, 2006*). Mechanistically, epigenetic regulations on DNA and histone proteins have been widely studied in the context of learning and memory (*Day and Sweatt, 2011*; *Levenson and Sweatt, 2005*). However, the roles of chemical modifications on mRNA in learning and memory have only become appreciated in recent years (*Fischer et al., 2007*; *Levenson et al., 2004*; *Kouzarides, 2007*). Especially, *N6*-methyladenosine (m6A), a well-studied form of mRNA methylation, has been shown to regulate synaptic plasticity as well as learning and memory (*Merkurjev et al., 2018*; *Zhuang et al., 2023*; *Shi et al., 2018*).

*N4*-acetylcytidine (ac4C) is the only known form of RNA acetylation in eukaryotes (*Boccaletto et al., 2018*). Initially, ac4C was identified in the transfer RNA (tRNA) anticodon (*Stern and Schulman, 1978*; *Oashi et al., 1972*; *Kowalski et al., 1971*; *Kruppa and Zachau, 1972*). Later studies revealed that ac4C also occurred in the 18S ribosomal RNA (rRNA) (*Ito et al., 2014a*). More recently, ac4C has been found in mRNA from mammalian cells, where it was reported to promote mRNA stability and translational efficiency (*Arango et al., 2018*; *Arango et al., 2022*). *N*-acetyltransferase 10 (NAT10) is the only known ac4C writer in mammalian cells (*Ito et al., 2014b*). NAT10-mediated mRNA acetylation has been reported to regulate tumorigenesis (*Liu et al., 2023a*; *Wang et al., 2022*), stem cell differentiation (*Hu et al., 2024*), cardiac remodeling (*Shi et al., 2023*), germ cell maturation (*Jiang et al., 2023*), and pain signaling in the spinal cord (*Zhang et al., 2023b*; *Xu et al., 2023*). However, the function and regulatory mechanisms of mRNA acetylation in the brain remain poorly understood.

The hippocampus plays an important role in learning and memory (*Jarrard, 1993*). Pyramidal neurons in the CA3 region of the hippocampus establish direct projection to CA1 pyramidal neurons through the Schaffer collateral (SC) pathway, known as the SC-CA1 synapse. The glutamatergic transmission at SC-CA1 synapses was found to be enhanced following training in the Morris water maze (MWM) (*Moser et al., 1994*), a behavioral paradigm for learning and memory (*Morris, 1984*). Long-term potentiation (LTP), a widely studied form of synaptic plasticity in hippocampus, is most closely

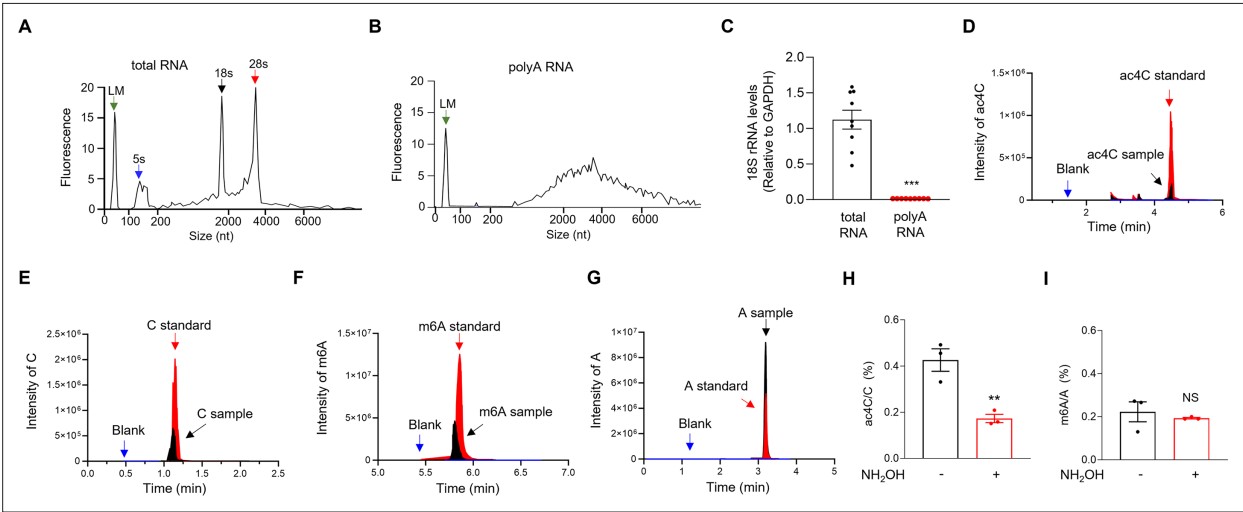

**Figure 1.** The presence of ac4C modification in PolyA RNA of mouse hippocampus. (**A, B**) Verification of the purity of PolyA RNA through the bioanalyzer profiling. The x-axis represents the size of nucleotides, and the y-axis represents the fluorescent signal from the bioanalyzer. LM, Lower Marker. (**A**) Total RNA. (**B**) PolyA RNA. (**C**) Diminished levels of 18S rRNA in the polyA RNA samples, compared with total RNA samples, which was revealed by RT-qPCR analysis. The levels of 18S rRNA were normalized by that of *Gapdh*. ***p < 0.001, two-tailed *t*-test, n = 6 biological replicates. The chromatograms of blank, standard, and testing samples of ac4C (**D**), C (**E**), m6A (**F**), and A (**G**) in the liquid chromatography–tandem mass spectrometry (LC–MS/MS) analysis. Shown are the representative peaks of ac4C, C, m6A, and A identified in the polyA RNA samples purified from the HOM of mouse hippocampus. (**H**) Reduction of ac4C levels by the chemical deacetylation protocol. The stoichiometry levels of ac4C in polyA RNA from control and $NH_2OH$-treated samples were quantified. **p = 0.008, two-tailed *t*-test, n = 3 biological replicates. (**I**) Unchanged m6A levels by the chemical deacetylation protocol. The stoichiometry levels of m6A in polyA RNA from control and $NH_2OH$-treated samples were quantified. NS, not significant, two-tailed *t*-test, n = 3 biological replicates. Quantification data are expressed as mean ± SEM.

linked to learning and memory (*Bliss and Collingridge, 1993*). Several studies showed that mRNAs and ribosomes were present in the synaptosomes, contributing to protein synthesis at synapses (*Biever et al., 2020*). In principle, neural activities can regulate local protein synthesis through post-transcriptional modification of synaptic mRNAs (*Goldie and Cairns, 2012*). However, whether synaptic mRNAs are subjected to NAT10-mediated acetylation and whether this process contributes to synaptic plasticity and memory remains elusive.

In this study, we conducted ac4C dot-blots, liquid chromatography–tandem mass spectrometry (LC–MS/MS), and acRIP-seq to study the ac4C modification of mRNA in mouse hippocampus during and after training in the MWM. We found that ac4C modification of mRNA was increased in the synaptosomes (SYN), rather than in the homogenates (HOM), following memory, and subsequently returned to baseline after memory forgetting. We further demonstrate that NAT10, the ac4C writer, in mouse hippocampus is important for memory through regulating memory-related mRNAs, proteins, and ultimately synaptic plasticity. Collectively, these results demonstrate the importance of mRNA acetylation in memory. The dataset of ac4C epitranscriptome from mouse hippocampus is freely accessible at the website (http://ac4catlas.com/).

## Results

### Presence of ac4C modification of polyA RNA in mouse hippocampus

Recent studies raised the debate about the presence of ac4C mRNA from mammalian cells (*Beiki et al., 2024*; *Georgeson and Schwartz, 2024*). We first investigate whether ac4C is a detectable modification of mRNA from mouse hippocampus. To this end, we purified polyadenylated (polyA) RNA from HOM of mouse hippocampus and determined their ac4C levels using LC–MS/MS. The purity of polyA RNA was verified by bioanalyzer analysis (*Figure 1A, B*) and dramatic reduction of 18S rRNA relative to total RNA in the analysis of reverse transcription followed by quantitative PCR (RT-qPCR) (*Figure 1C*). An ac4C standard curve was generated to calculate the stoichiometry levels of ac4C in polyA RNA samples. Both the ac4C (or C) and m6A (or A) peaks can be detected by the chromatograms in the standard samples but not in the blank solution (*Figure 1D–G*). The polyA RNA purified from the HOM of mouse hippocampus showed ac4C peaks with a stoichiometry level of $0.426 \pm 0.049\%$ ($n = 3$ biological replicates) (*Figure 1H*). Likewise, the polyA RNA purified from the HOM of mouse hippocampus showed m6A peaks with a stoichiometry level of $0.223 \pm 0.046\%$ ($n = 3$ biological replicates) (*Figure 1I*), which is consistent with the previous study from mouse brain (*Li et al., 2023*). When the polyA RNA samples were incubated with 50 mM $NH_2OH$ for 1 hr before LC–MS/MS, a protocol of chemical deacetylation (*Sinclair et al., 2017*), the stoichiometry levels of ac4C were significantly reduced (*Figure 1H*), indicating the validity of the LC–MS/MS analysis. By contrast, the stoichiometry levels of m6A in polyA RNA samples remained unchanged after $NH_2OH$ treatment (*Figure 1I*), supporting the specificity of the deacetylation protocol. Altogether, these results demonstrate the presence of ac4C modification in polyA RNA from mouse hippocampus. Given that polyA RNA consists of mRNA and long noncoding RNA (lncRNA) (*Yu et al., 2020*), we further performed acRIP-seq to analyze the ac4C mRNA in mouse hippocampus.

### Characterization of ac4C mRNA in the HOM of mouse hippocampus after memory

We used the behavioral protocol of MWM (*Vorhees and Williams, 2006*) to study how memory regulates mRNA acetylation in mouse hippocampus. The control mice were placed in the MWM with a visible platform once a day for 2 days and thus could reach the visible platform in the target region without training (*Figure 2—figure supplement 1*). However, the control mice could not remember the target region during the probe test at day 6 (*Figure 2—figure supplement 1*). In contrast, the memory mice were subjected to training in the MWM with a hidden platform (*Figure 2—figure supplement 1*). The memory mice took 5 days of training to remember the target region during the probe test at day 6 (i.e., post-training day 1) (*Figure 2—figure supplement 1*), indicative of memory. Consistent with the previous study (*de Hoz et al., 2004*), the memory mice could not remember the target region during the probe test at day 20 after staying in the home cages for 2 weeks (i.e., post-training day 15) (*Figure 2—figure supplement 1*), indicating that memories were naturally forgotten (*Figure 2—figure supplement 1*).

We next performed acRIP-seq (*Arango et al., 2018*) to identify ac4C mRNA in the HOM of mouse hippocampus. Two libraries of RNA (acRIP and input) were subjected to the next-generation sequencing (NGS) for each sample (*Supplementary file 1A*). The acRIP-seq reads were mapped to mouse hippocampal transcriptome to identify ac4C enriched regions relative to input, and the fold enrichment (FE) of an ac4C peak was defined as acRIP/input (*Figure 2—figure supplement 2*). An ac4C peak was considered positive when its FE >2 and adjusted p-values <0.05 (*Zhang et al., 2023a*).

We performed acRIP-seq for three HOM replicates from control and memory mice 1 hr after probe tests in the MWM (i.e., at day 6). The ac4C peaks and ac4C mRNAs identified in three biological replicates of control and memory mice were largely overlapped (*Figure 2—figure supplement 2*). These results indicate the consistency of the acRIP-seq for mouse hippocampus. The ac4C peak that was identified from at least two of the three biological replicates was considered reliable. Only mRNAs corresponding to these reliable ac4C peaks are defined as reliably ac4C mRNAs. We identified 14,516 ac4C peaks and 8303 ac4C mRNAs from the HOM of control mice (*Figure 2—figure supplement 2 and Supplementary file 1B*), and 13,348 ac4C peaks and 8107 ac4C mRNAs from the HOM of memory mice (*Figure 2—figure supplement 2* and *Supplementary file 1B*). Moreover, the acRIP-seq revealed the same consensus motif AGCAGCTG in the ac4C peak regions from three biological replicates of control and memory mice (*Figure 2—figure supplement 2*).

When we compared the transcripts with ac4C modifications to the entire transcriptome, we found that 54.1 ± 2.62% (*n* = 3 biological replicates) of the mRNAs in the mouse hippocampus were modified by ac4C. In line with the previous study from HeLa cells (*Arango et al., 2018*), metagene profile analysis revealed that most ac4C peaks were located within the coding sequence (CDS) of mRNAs in mouse hippocampus, with lower density in the 5′ and 3′ untranslated regions (*Figure 2—figure supplement 2*). To further assess the reproducibility of our ac4C detection, we performed Upset plot analysis, which revealed 4494 peaks common to all three biological replicates in control mice and 4110 peaks in memory mice (*Figure 2—figure supplement 2*). The numbers of ac4C peaks per acetylated mRNA were comparable across three biological replicates and between control and memory mice (*Figure 2—figure supplement 2*). Likewise, the ac4C dot-blots of polyA RNA, and the cumulative distribution of ac4C mRNAs in the HOM were similar between the control and memory mice (*Figure 2—figure supplement 2*). The cumulative distribution of mRNA counts in the HOM that were revealed by the NGS of input RNA was also similar between control and memory mice (*Figure 2—figure supplement 2*). Altogether, these results suggest that the total levels of ac4C mRNA in the HOM of mouse hippocampus may not be altered by memory.

## Dynamic change of ac4C polyA RNA in the SYN of mouse hippocampus during memory

We next purified the SYN fraction from mouse hippocampus, enriched for synaptic proteins (PSD95 and synapsin 1) and depleted of cytoplasm (the S2 fraction)-restricted protein (α-tubulin) (*Figure 2A*). We then analyzed the levels of ac4C polyA RNA purified from the SYN with the ac4C dot-blots. The control and memory mice at day 1 (i.e., training phase day 1 for memory mice) showed similar levels of ac4C polyA RNA in the SYN (*Figure 2B*). Strikingly, the levels of ac4C polyA RNA showed a significant increase in the SYN of memory mice at day 6, compared with controls (*Figure 2C*). However, the levels of ac4C polyA RNA returned to control levels in the SYN of memory mice at day 20 (*Figure 2D*). These results suggest that acetylation of polyA RNA was upregulated in the SYN of mouse hippocampus after memory formation but returned to control levels after memory forgetting. The dynamic change of ac4C RNA in the SYN of mouse hippocampus during memory was verified by the assay of LC–MS/MS (*Figure 2E*).

## Characterization of ac4C mRNA in the SYN of mouse hippocampus

Given that the amount of RNA that could be purified from SYN (200 ng per mouse hippocampus) is much less than that from HOM (50 μg per mouse hippocampus), we used the protocol of low-input acRIP-seq to identify ac4C mRNAs in the SYN (see details in the Methods). Due to the limited yield of SYN RNA, we purified the SYN RNA from the hippocampus of 10 adult mice and combined them into one sample. Six total samples from control and memory mice at days 1, 6, and 20 were subjected to low-input acRIP-seq. Two libraries (acRIP and input) were sequenced for each sample (*Supplementary file 1A*). To increase the data reliability for low-input acRIP-seq, transcripts that were identified in our

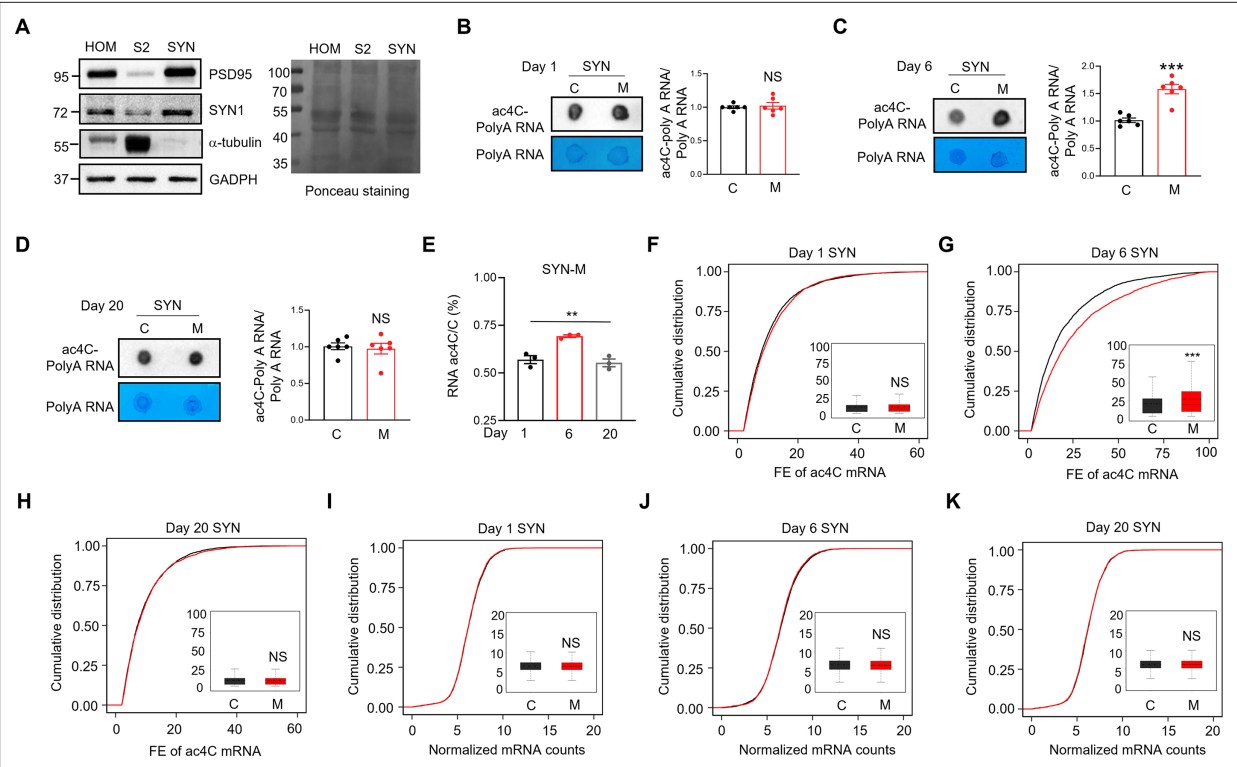

**Figure 2.** Increased mRNA acetylation in the SYN of mouse hippocampus after memory. (**A**) The purified synaptosomes (SYN) of mouse hippocampus expressed high levels of PSD95 and synapsin 1 (SYN1), two synaptic protein markers, but very low levels of α-tubulin, a cytoplasmic protein marker. The homogenates (HOM), cytoplasm (S2), and SYN of mouse hippocampus were subjected to western blots with the indicated antibodies. Left panel, representative images of western blots; right panel, Ponceau staining to show equal protein loading. (**B**) Acetylation of polyA RNA was not altered in SYN of mouse hippocampus at day 1. The polyA RNA purified from the SYN of control and memory mice at day 1 after Morris water maze (MWM) training was subjected to ac4C dot-blots and methylene blue staining. Left, the representative images; right, quantification data. NS, not significant, two-tailed *t*-test, *n* = 6 biological replicates. Data were normalized by controls. (**C**) Acetylation of polyA RNA was increased in the SYN of mouse hippocampus at day 6. The polyA RNA purified from the SYN of control and memory mice at day 6 after MWM training was subjected to ac4C dot-blots and methylene blue staining. Left, the representative images; right, quantification data. ***p < 0.001, two-tailed *t*-test, *n* = 6 biological replicates. Data were normalized by controls. (**D**) Acetylation of polyA RNA was not altered in the SYN of mouse hippocampus at day 20. The polyA RNA purified from the SYN of control and memory mice at day 20 after MWM training was subjected to ac4C dot-blots and methylene blue staining. Left, the representative images; right, quantification data. NS, not significant, two-tailed *t*-test, *n* = 6 biological replicates. Data were normalized by controls. (**E**) Acetylation of RNA in the SYN was increased during memory but returned to normal levels after forgetting. The stoichiometry levels of ac4C of total RNA purified from the SYN of memory (M) mice were shown at days 1, 6, and 20 after MWM training. **p = 0.0023, one-way-ANOVA, *n* = 3 biological replicates. (**F–H**) Acetylation of mRNA was increased after memory but returned to normal levels after forgetting in the SYN of mouse hippocampus. Shown are cumulative distribution curves of FE of ac4C mRNA purified from the SYN of control (C) and memory (M) mice at days 1 (**F**), 6 (**G**), and 20 (**H**) after MWM training. The FE of ac4C mRNA was quantified in the quartile boxplots, where the solid line in the box is the median and the dashed line is the mean, where the maximum and minimum values are identified. NS, not significant, ***adj-p = 9.9814e−13, Kolmogorov–Smirnov test. (**I–K**) The total mRNA levels were not significantly altered in the SYN of mouse hippocampus during memory. Shown are cumulative distribution curves of normalized mRNA counts for SYN in control (C) and memory (M) mice at days 1 (**I**), 6 (**J**), and 20 (**K**) after MWM training. The normalized mRNA counts were quantified in the quartile boxplots, where the solid line in the box is the median and the dashed line is the mean, where the maximum and minimum values are identified. NS, not significant, Kolmogorov–Smirnov test. C, control; M, memory. FE, fold enrichment. Quantification data are expressed as mean ± SEM.

The online version of this article includes the following source data and figure supplement(s) for figure 2:

**Source data 1.** PDF file containing original membranes corresponding to *Figure 2*, panel A.

**Source data 2.** Original files for western blot analysis displayed in *Figure 2A*.

**Figure supplement 1.** Protocols of Morris water maze (MWM) for control and memory mice.

**Figure supplement 2.** No significant change of ac4C mRNA in the homogenates of mouse hippocampus after memory.

own sequencing and at least two of the three synaptosome databases (*Cajigas et al., 2012*; *Hafner et al., 2019*; *Niu et al., 2023*) were considered reliable and included in the analysis.

The transcripts and ac4C mRNAs identified from the SYN of control and memory mice at days 1, 6, and 20 were largely overlapped (*Figure 3—figure supplement 1*; *Supplementary file 1C*). We found that 58.61 ± 1.39% of the mRNAs in the SYN from control mouse hippocampus were modified by ac4C. On average, every acetylated transcript had 1.77 ± 0.05 ac4C peaks in the SYN from control mouse hippocampus. Moreover, the low-input acRIP-seq revealed the same consensus motif AGCAGCTG in the ac4C peak regions of SYN mRNA from control and memory mice at days 1, 6, and 20 (*Figure 3—figure supplement 1*). In line with the results from HOM, most ac4C peaks were identified in CDS of SYN mRNA from both control and memory mice (*Figure 3—figure supplement 1*). These results indicate the consistency of the low-input acRIP-seq for SYN mRNA.

## Dynamic change of ac4C mRNA in the SYN of mouse hippocampus during memory

The cumulative distribution of ac4C mRNAs in the SYN did not differ between control and memory mice at day 1 (*Figure 2F*). In contrast, the cumulative distribution curve of ac4C mRNAs in the SYN of memory mice had a significant rightward shift at day 6, compared with control mice (*Figure 2G*). Intriguingly, the cumulative distribution of ac4C mRNAs in the SYN of memory mice became indistinguishable from that of control mice at day 20 (*Figure 2H*). The cumulative distribution of mRNA counts in the SYN was similar between control and memory mice at days 1, 6, and 20 (*Figure 2I–K*). These results suggest that acetylation of mRNA rather than the total levels of mRNA was increased in the SYN after memory, and mRNA acetylation returned to control levels after memory forgetting, supporting the dynamic change of acetylation of polyA RNA in the SYN during memory (*Figure 2B–E*).

To further validate the distinct molecular characteristics of the 'memory-day6' group, we performed principal component analysis based on global ac4C modification profiles. The first two principal components accounted for 38% and 26% of the total variance, respectively. The 'memory-day6' samples clustered distinctly from the other five groups, confirming a unique epitranscriptomic signature associated with memory (*Figure 3—figure supplement 1*). To identify the memory-induced synaptic ac4C (MISA) mRNA, we analyzed the fold change (FC) of ac4C mRNA in the SYN of memory versus control mice (L/C) at days 1, 6, and 20 (*Figure 3—figure supplement 1*). Here, we analyzed the FE of all ac4C peaks to better reflect the acetylation level of individual mRNA. Acetylation of MISA mRNA should be upregulated after memory and return to control levels after memory forgetting. In other words, the ac4C of MISA mRNA should be upregulated specifically at day 6 (the cutoff value of upregulation is FC >2) but not at days 1 or 20. Per these criteria, we identified 1237 MISA mRNAs whose ac4C levels were increased at day 6 rather than at days 1 or 20 in memory mice, compared with controls (*Figure 3—figure supplement 1*; *Supplementary file 1D*). In contrast, the transcription levels of MISA mRNAs were similar at days 1, 6, and 20 between control and memory mice (*Figure 3—figure supplement 1*; *Supplementary file 1D*).

## A representative MISA mRNA for *Arc*

Gene ontology (GO) analysis indicated that MISA mRNAs were significantly enriched in the biological processes (BPs) related to synaptic plasticity and learning or memory (*Figure 3A*; *Supplementary file 1D*). Protein–protein interaction (PPI) network analysis of MISA mRNAs involved in learning or memory revealed several core genes such as activity-regulated cytoskeleton associated protein (*Arc*), glutamate receptor NMDA type subunit 1 (*Grin1*), and presenilin 1 (*Psen1*) (*Figure 3A, B*). ARC protein can be transported into dendrites and translated near activated postsynaptic sites after neural activities (*Steward et al., 1998*; *Lyford et al., 1995*). Different from other immediately early genes, *Arc* underwent repetitive cycles of transcription and local translation, which is critical for memory consolidation (*Das et al., 2023*; *Steward et al., 2014*). Although *Arc* is a promising 'master regulator' of protein synthesis-dependent forms of memory, the mechanisms underlying the regulation of local translation of *Arc* in the SYN remained to be elucidated (*Bramham et al., 2010*). Analysis of the acRIP-seq data indicates that memory mice have elevated ac4C *Arc* mRNA in the SYN at day 6 rather than days 1 or 20, and similar levels of ac4C *Arc* mRNA in the HOM at day 6, compared with control mice (*Figure 3C*). The genome browser tracks for the ac4C of all MISA mRNAs were accessible at the ac4C

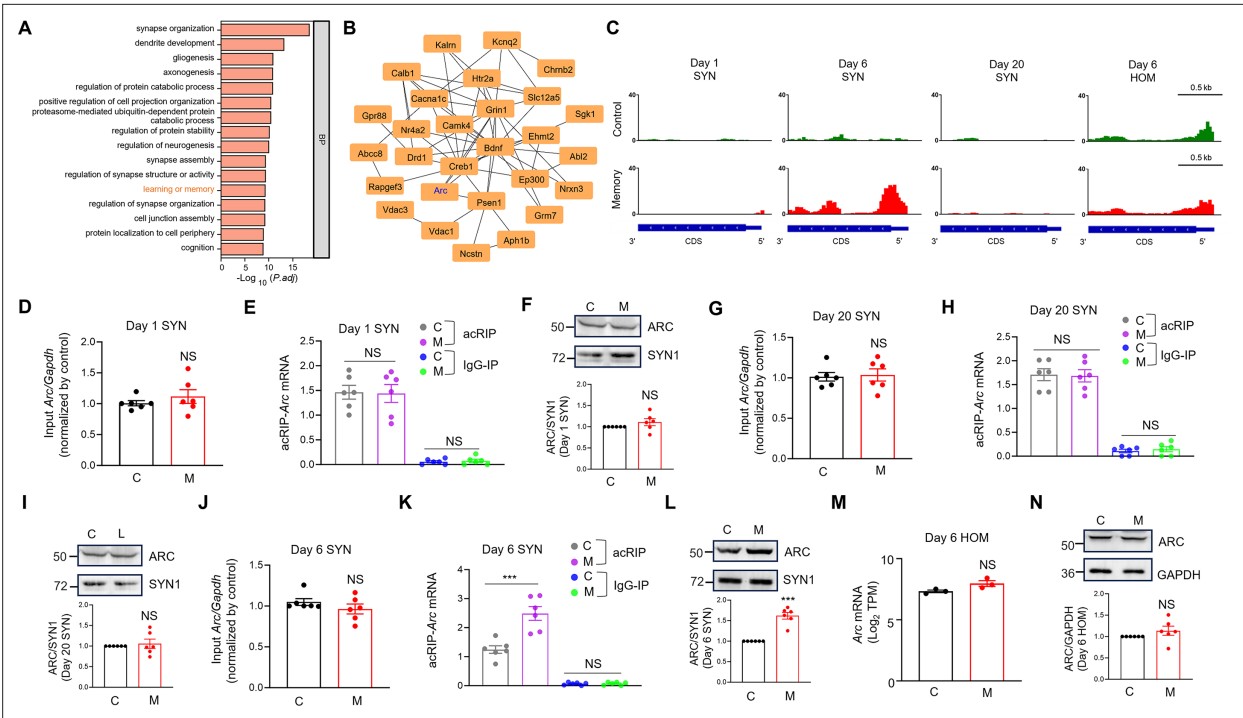

**Figure 3.** A representative MISA mRNA for *Arc*. (**A**) The top 16 biological process (BP) in which MISA mRNAs are significantly overrepresented. The *x*-axis represents the negative logarithm of the adjusted p-value, reflecting the significance of the enrichment, while the *y*-axis shows the BPs. (**B**) The protein–protein interaction (PPI) network of MISA mRNA implicated in the BP of learning or memory. $p < 1.0e-16$, Hypergeometric Test. (**C**) Increased ac4C modification of *Arc* in the SYN rather than in the HOM at day 6. Also note the unaltered ac4C *Arc* mRNA at day 1 or 20 in the SYN. Shown are the IGV maps of the ac4C peaks from the 3' UTR to the 5' UTR of *Arc* mRNA. The ac4C peaks were represented by acRIP/input (see details in the Method). *Arc* mRNA levels from the input of SYN were not altered at days 1 (**D**) or 20 (**G**) after Morris water maze (MWM) training. The RNA purified from the SYN was subjected to the RT-qPCR analysis. NS, not significant, two-tailed *t*-test, *n* = 6 biological replicates. The levels of *Arc* mRNA were normalized by that of *Gapdh* in each sample, and then the ratio of *Arc* to *Gapdh* was normalized by controls. The ac4C modification of *Arc* mRNA was not altered in the SYN at days 1 (**E**) or 20 (**H**) after MWM training. The RNA purified through the acRIP and IgG-IP products of SYN was subjected to the RT-qPCR analysis. NS, not significant, two-way ANOVA followed by Tukey's multiple comparison test, *n* = 6 biological replicates. Data were normalized by input. The protein levels of ARC did not change in the SYN at days 1 (**F**) or 20 (**I**) after MWM training. Top, representative western blot images; bottom, quantification data. NS, not significant, two-tailed *t*-test, *n* = 6 biological replicates. The protein levels of ARC were normalized by that of SYN1 in each sample, and then the ratio of ARC to SYN1 was normalized by controls. (**J**) *Arc* mRNA levels from the input of SYN were not altered at day 6 after MWM training. The RNA purified from the SYN was subjected to the RT-qPCR analysis. NS, not significant, two-tailed *t*-test, *n* = 6 biological replicates. The levels of *Arc* mRNA were normalized by that of *Gapdh* in each sample, and then the ratio of *Arc* to *Gapdh* was normalized by controls. (**K**) The ac4C modification of *Arc* mRNA was significantly increased in the SYN at day 6 after MWM training. The RNA purified from the acRIP and IgG-IP products of SYN was subjected to the RT-qPCR analysis. ***p < 0.001, two-way-ANOVA followed by Tukey's multiple comparison test, *n* = 6 biological replicates. Data were normalized by input. (**L**) The protein levels of ARC were significantly increased in the SYN at day 6 after MWM training. Top, representative western blot images; bottom, quantification data. ***p < 0.001, two-tailed *t*-test, *n* = 6 biological replicates. The protein levels of ARC were normalized by that of SYN1 in each sample, and then the ratio of ARC to SYN1 was normalized by controls. (**M**) *Arc* mRNA levels from the HOM were not altered at day 6 after MWM training. The *Arc* mRNA levels were revealed by the log2TPM from the next-generation sequencing of the HOM RNA. NS, not significant, two-tailed *t*-test, *n* = 3 biological replicates. (**N**) The protein levels of ARC did not change in the HOM at day 6 after MWM training. Top, representative western blot images; bottom, quantification data. NS, not significant, two-tailed *t*-test, *n* = 6 biological replicates. The protein levels of ARC were normalized by that of GAPDH in each sample, and then the ratio of ARC to GAPDH was normalized by controls. C, control; M, memory. Quantification data are expressed as mean ± SEM.

The online version of this article includes the following source data and figure supplement(s) for figure 3:

**Source data 1.** PDF file containing original membranes corresponding to *Figure 3*, panel F, I, L, N.

**Source data 2.** Original files for western blot analysis displayed in *Figure 3*.

**Figure supplement 1.** Memory-induced synaptic ac4C mRNAs (MISA) in mouse hippocampus.

website (http://ac4catlas.com/) to show altered ac4C in the SYN, and similar lack of change for those same RNAs in the HOM at day 6.

Next, we used an orthogonal method to verify the dynamic change of ac4C *Arc* mRNA in the SYN during memory. To this end, we performed RT-qPCR analysis for *Arc* mRNA from the input and acRIP samples from the SYN. The levels of input and acRIP-*Arc* mRNA were similar in the SYN of control and memory mice at days 1 and 20 (*Figure 3D, E, G, H*). Likewise, the protein levels of ARC were also comparable in the SYN of control and memory mice at days 1 and 20 (*Figure 3F, I*). Although the levels of input *Arc* mRNA were not altered in the SYN of memory mice at day 6 (*Figure 3J*), the levels of acRIP-*Arc* mRNA and ARC proteins were significantly increased in the SYN of memory mice at day 6, compared with controls (*Figure 3K, L*). In contrast, the levels of *Arc* mRNA and ARC proteins were similar in the HOM of control and memory mice at day 6 (*Figure 3M, N*). Note that *Arc* mRNA levels in the IgG-IP samples were much lower than that in the acRIP samples (*Figure 3E, H, K*), indicating the specificity of RT-qPCR for acetylated *Arc* mRNA. Altogether, the results demonstrate that acetylation of *Arc* mRNA is specifically increased in the SYN after memory but returned to control levels after memory forgetting.

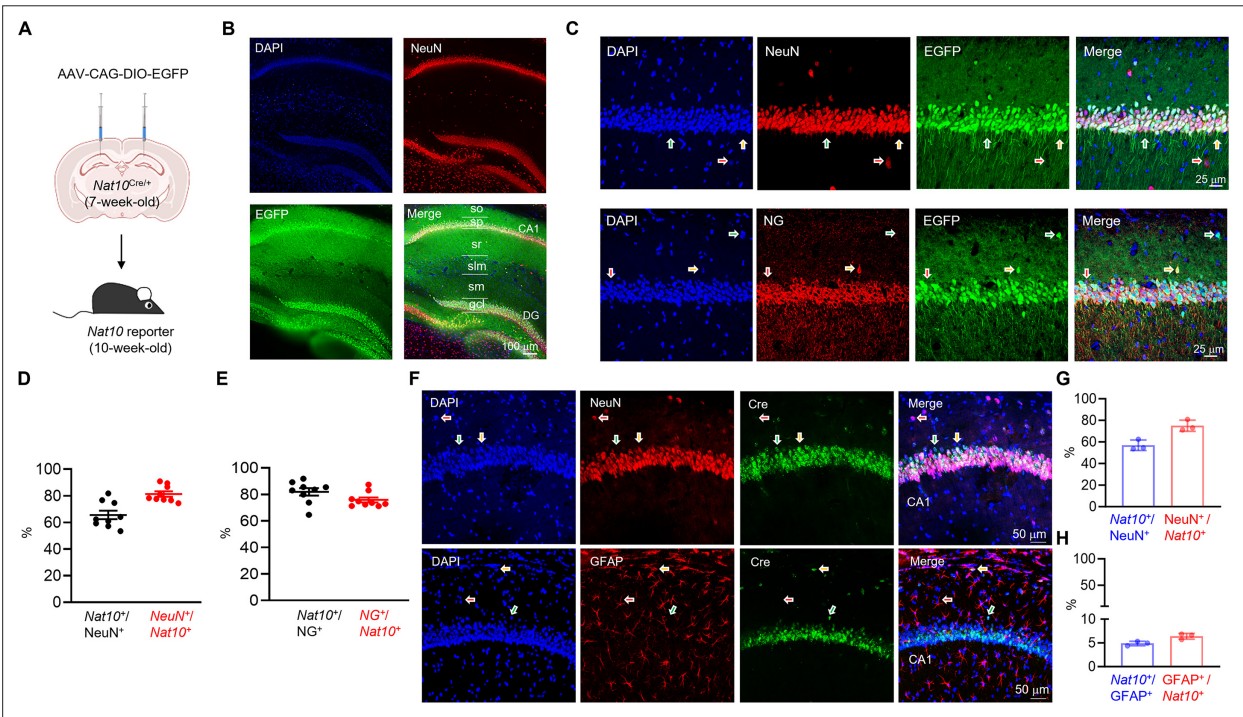

**Figure 4.** Expression of *Nat10* in mature neurons of mouse hippocampus. (**A**) Experimental design. Cre-reporting AAV was injected into the dorsal hippocampus of 7-week-old *Nat10*^Cre/+^ mice. Three weeks after AAV injection, the fluorescent protein EGFP was analyzed in the *Nat10* reporter mice. (**B**) Co-localization of EGFP with NeuN in the hippocampus of *Nat10* reporter mice. The hippocampal slices from *Nat10* reporter mice were immunostained with anti-NeuN antibodies. So, stratum oriens; sp, stratum pyramidale; sr, stratum radiatum; slm, stratum lacunosum-moleculare; sm, stratum moleculare; gcl, granule cell layer; DG, dentate gyrus. Scale bar, 100 µm. (**C**) Co-localization of EGFP and NeuN (top) or NG (bottom) in the CA1 region of *Nat10* reporter mice. The hippocampal slices from *Nat10* reporter mice were immunostained with anti-NeuN or anti-NG antibodies. The yellow arrows indicate a neuron (top) or excitatory pyramidal neuron (bottom) expressing *Nat10*, whereas the red arrows represent a neuron (top) or excitatory pyramidal neuron (bottom) not expressing *Nat10*. The green arrows indicate non-neuronal cells that express *Nat10*. Scale bar, 25 µm. (**D**) Quantification of the percentage of *Nat10*-expressing neurons (*Nat10*^+^/NeuN^+^ = 65.61 ± 3.245%) and the percentage of neurons in *Nat10*-positive cells (NeuN^+^/*Nat10*^+^ = 81.31 ± 2.029%) in panel C. n = 9 slices from 3 mice. (**E**) Quantification of the percentage of *Nat10*-expressing excitatory neurons (*Nat10*^+^/NG^+^=81.87 ± 2.765%) and the percentage of excitatory neurons in *Nat10*-positive cells (NG^+^/*Nat10*^+^ = 75.82 ± 1.882%) in panel C. n = 9 slices from 3 mice. (**F**) Co-localization of Cre with NeuN (top) or GFAP (bottom) in the hippocampus of *Nat10*^Cre/+^ mice. Hippocampal sections from *Nat10*^Cre/+^ mice were immunostained with anti-NeuN or anti-GFAP antibodies. The yellow arrows indicate a neuron (top) or astrocyte (bottom) expressing *Nat10*, whereas the red arrows represent a neuron (top) or astrocyte (bottom) not expressing *Nat10*. The green arrows indicate *Nat10*-positive cells not expressing NeuN (top) or GFAP (bottom). Scale bar, 50 µm. (**G**) Quantification of the percentage of *Nat10*-expressing neurons (*Nat10*^+^/NeuN^+^ = 56.93 ± 2.795%) and the percentage of neurons in *Nat10*-positive cells (NeuN^+^/*Nat10*^+^ = 75.07 ± 2.968%) in panel F. n = 3 slices from 3 mice. (**H**) Quantification of the percentage of *Nat10*-expressing astrocytes (*Nat10*^+^/GFAP^+^=4.88 ± 0.273%) and the percentage of astrocytes in *Nat10*-positive cells (GFAP^+^/*Nat10*^+^ = 6.39 ± 0.346%) in panel F. n = 3 slices from 3 mice.

## Expression of *Nat10* in the mature neurons of mouse hippocampus

NAT10 is the only known ac4C writer (*Ito et al., 2014b*). Since the commercially available NAT10 antibodies were not suitable for immunostaining in the hippocampal slices, we used the strategy of genetic labeling (*Yu et al., 2019*; *Madisen et al., 2010*) to investigate the expression pattern of *Nat10*. To this end, a heterozygous knock-in mouse line (*Nat10*$^{Cre/+}$) that expressed the Cre recombinase right after the start codon (ATG) of *Nat10* gene was generated using the CRISPR/Cas9 techniques (see details in the Method). Since the Cre recombinase and *Nat10* share the same regulatory elements for gene expression in the *Nat10*$^{Cre/+}$ mice, the *Nat10* expression pattern can be revealed through the Cre-reporting adeno-associated virus (AAV) that express EGFP in a Cre-dependent manner. In this way, the *Nat10*-positive cells would express EGFP in the *Nat10* reporter mice (*Figure 4A*).

The dorsal hippocampus integrates multiple synaptic inputs from other brain regions and is critical for learning and memory (*Biane et al., 2023*; *Fanselow and Dong, 2010*; *Moser et al., 1993*). To study the expression pattern of *Nat10* in the dorsal hippocampus of adult mice, we injected Cre-reporting AAV into the dorsal hippocampus of 7-week-old *Nat10*$^{Cre/+}$ mice, and the fluorescent protein EGFP was analyzed 3 weeks after AAV injection. Analysis of the EGFP-positive cells revealed that *Nat10* was mainly expressed in the pyramidal cell layer (sp) of the CA1 region and granule cell layer (gcl) of dentate gyrus (DG) (*Figure 4B*). Around 80% of *Nat10*-expressing cells are mature neurons, evidenced by the overlay between EGFP and NeuN, a protein marker for mature neurons (*Figure 4C, D*). Note that about 20% of *Nat10*-positive cells are non-neuronal and probably glial cells. More-over, about 75% of *Nat10*-positive cells are excitatory pyramidal neurons in the CA1 region of mouse hippocampus, indicated by the co-localization of EGFP with Neurogranin (NG), a protein marker for excitatory pyramidal neurons (*Figure 4C, E*).

To validate the expression pattern of *Nat10*, we observed Cre signals in the *Nat10*$^{Cre/+}$ mice using immunofluorescence. We found that most of the Cre-positive cells were mature neurons within the CA1 region, with a small subset being GFAP-positive astrocytes (*Figure 4F–H*). Collectively, these results indicate that *Nat10* is mainly expressed in mature neurons, with limited expression in glia cells.

## Increase of NAT10 proteins in the SYN after memory

The above-mentioned data showed that the levels of ac4C mRNA were increased in the SYN of mouse hippocampus after memory. Next, we aim to investigate whether the protein levels of NAT10, the only known ac4C writer (*Ito et al., 2014b*), are also regulated by memory. Toward this aim, we purified the SYN fraction from the hippocampus of control and memory mice at days 1, 6, and 20, and analyzed the protein levels of NAT10 by western blots (*Figure 5A*). The protein levels of NAT10 in the SYN were similar between control and memory mice at day 1 (*Figure 5B*). Interestingly, the protein levels of NAT10 were significantly increased in the SYN of memory mice at day 6 (*Figure 5C*), but returned to control levels at day 20 (*Figure 5D*). These results indicate that protein levels of NAT10 are increased in the SYN after memory but return to normal levels after memory forgetting (*Figure 5E*).

The protein levels of NAT10 were not altered in HOM of memory mice at day 6, compared to controls (*Figure 5F*), suggesting that memory did not upregulate the de novo protein synthesis of NAT10 in mouse hippocampus. To further explore the distribution of NAT10 proteins in other subcellular fractions, we purified the nucleus (N), cytoplasm (S2), and postsynaptic density (PSD) fractions from the hippocampus of control and memory mice at day 6 and analyzed the protein levels of NAT10 in different subcellular fractions (*Figure 5G*). The nuclear fraction was enriched for histone but was depleted of cytoplasmic protein α-tubulin or synaptic protein SYN1 and PSD95, and vice versa for the S2 and PSD fractions (*Figure 5H*). The protein levels of NAT10 in the nucleus did not change at day 6 after memory (*Figure 5I*). Strikingly, the NAT10 proteins were significantly reduced in the cytoplasm (S2 fraction) but increased in the PSD fraction at day 6 after memory (*Figure 5J and K*). These results suggest that NAT10 proteins might be relocated from the cytoplasm to synapses in mouse hippocampus after memory.

NAT10 belongs to the p300/CBP-associated factor (PCAF) subfamily of protein acetyltransferase (*Lau et al., 2000*). Next, we aim to investigate whether memory changed protein levels of PCAF in the subcellular fractions of mouse hippocampus. The protein levels of PCAF did not change in the nucleus, cytoplasm, or PSD fractions of memory mice at day 6, compared to controls (*Figure 5L–N*), indicating that memory may not alter the subcellular distribution of PCAF.

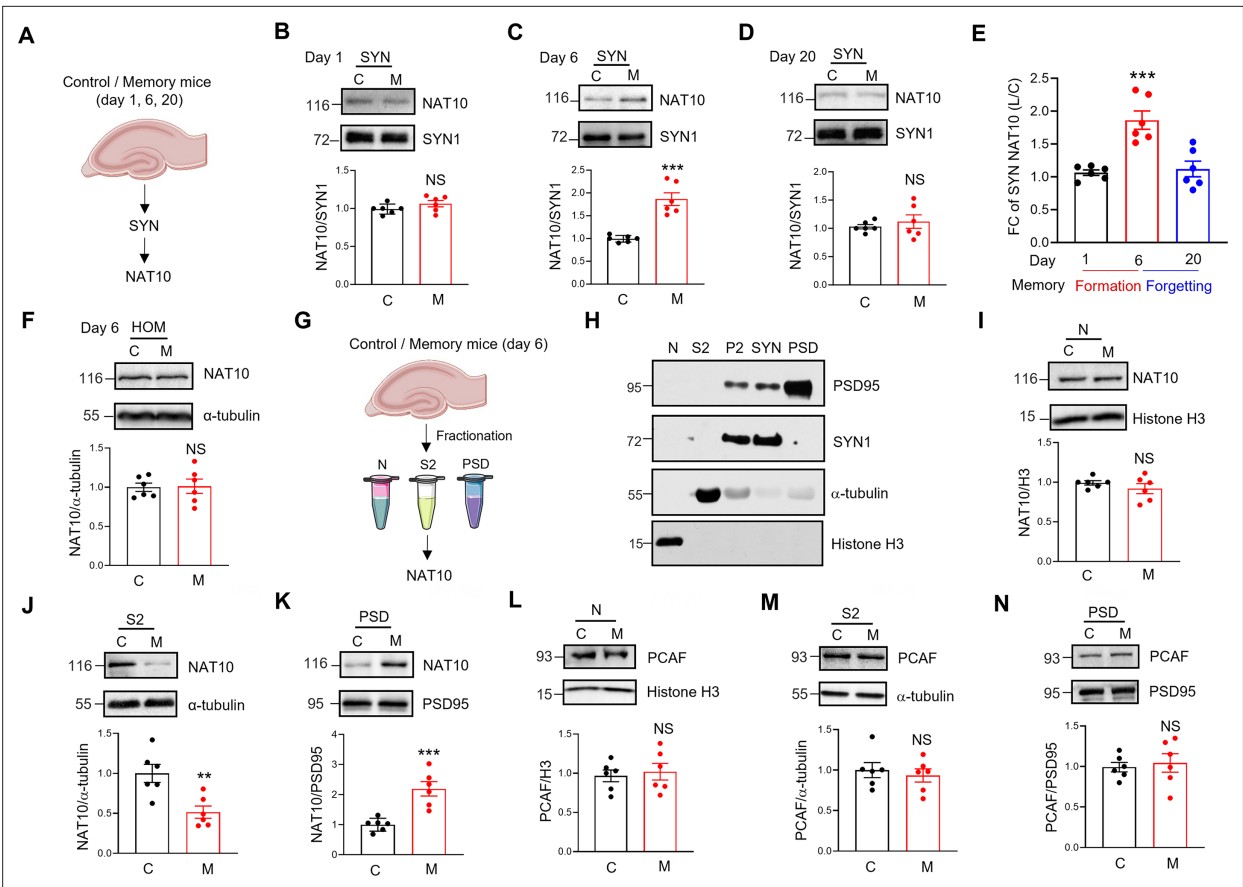

**Figure 5.** Increased NAT10 proteins in the SYN of mouse hippocampus after memory. (**A**) Experimental design. The SYN fractions were purified from the hippocampus of control and memory mice at days 1, 6, and 20. The NAT10 protein levels in the SYN were then assessed by western blots. (**B–D**) Protein levels of NAT10 were increased in the SYN after memory but returned to normal levels after forgetting. The SYN proteins from control and memory mice at days 1 (**B**), 6 (**C**), and 20 (**D**) were subjected to western blots with the indicated antibodies. The protein levels of NAT10 were normalized by that of SYN1 in each sample, and then the ratio of NAT10 to SYN1 was normalized by controls. Top, representative western blot images; bottom, quantification data. NS, not significant, ***p < 0.001, two-tailed *t*-test, n = 6 biological replicates. (**E**) The fold change (FC) of NAT10 proteins in panels B–D was quantified. ***p < 0.001, one-way-ANOVA, n = 6 biological replicates. (**F**) Protein levels of NAT10 were not altered in the HOM at day 6. The HOM proteins from control and memory mice at day 6 were subjected to western blots with the indicated antibodies. Top, representative western blot images; bottom, quantification data. NS, not significant, two-tailed *t*-test. Data were normalized by controls, n = 6 biological replicates at each condition. The protein levels of NAT10 were normalized by that of α-tubulin in each sample, and then the ratio of NAT10 to α-tubulin was normalized by controls. (**G**) Experimental design. Proteins were extracted from nucleus (**N**), cytoplasm (**S2**), and postsynaptic density (**PSD**) of control and memory mice hippocampus at day 6, and then NAT10 protein levels were assessed by western blots. (**H**) Expression of different protein markers in the nuclear (N), cytoplasmic (S2), P2 (Crude synaptosomes fraction), SYN, and PSD fractions of mouse hippocampus. Shown are representative images of western blots. Protein samples purified from different subcellular fractions of mouse hippocampus were subjected to western blots with the indicated antibodies. (**I**) Protein levels of NAT10 were not altered in the nuclear fraction of mouse hippocampus after memory. The nuclear protein samples from control and memory mice at day 6 were subjected to western blots with the indicated antibodies. Top, representative western blot images; bottom, quantification data. NS, not significant, two-tailed *t*-test, n = 4 biological replicates. The protein levels of NAT10 were normalized by that of Histone H3 proteins in each sample, and then the ratio of NAT10 to Histone H3 was normalized by controls. (**J**) Protein levels of NAT10 were significantly reduced in the S2 fraction of mouse hippocampus after memory. The cytoplasmic protein samples from control and memory mice at day 6 were subjected to western blots with the indicated antibodies. Top, representative western blot images; bottom, quantification data. **p = 0.0052, two-tailed *t*-test, n = 6 biological replicates. The protein levels of NAT10 were normalized by that of α-tubulin in each sample, and then the ratio of NAT10 to α-tubulin was normalized by controls. (**K**) Protein levels of NAT10 were significantly increased in the PSD fraction of mouse hippocampus after memory. The PSD protein samples from control and memory mice at day 6 were subjected to western blots with the indicated antibodies. Top, representative western blot images; bottom, quantification data. ***p = 0.0008, two-tailed *t*-test. n = 6 biological replicates. The protein levels of NAT10 were normalized by that of PSD95 in each sample, and then the ratio of NAT10 to PSD95 was normalized by controls. Protein levels of p300/CBP-associated factor (PCAF) were not altered in the nuclear (**L**), S2 (**M**), or PSD fraction (**N**) of mouse hippocampus after memory. The protein samples purified from the different subcellular fractions of the hippocampus of control and memory mice at day 6 were subjected to western blots with the indicated antibodies. Top, representative western blot images; bottom, quantification data. NS, not significant, two-tailed *t*-test, n = 6 biological replicates. The protein levels were normalized by

*Figure 5 continued on next page*

*Figure 5 continued*

the protein markers in different subcellular fractions and then were normalized by controls. C, control; M, memory. Quantification data are expressed as mean ± SEM.

The online version of this article includes the following source data and figure supplement(s) for figure 5:

**Source data 1.** PDF file containing original membranes corresponding to *Figure 5*.

**Source data 2.** Original files for western blot analysis displayed in *Figure 5*.

**Figure supplement 1.** Increase of NAT10 proteins and ac4C modification of *Arc* mRNA in the SYN of mouse hippocampal neurons and slices after cLTP stimulation.

**Figure supplement 1—source data 1.** PDF file containing original membranes corresponding to *Figure 5—figure supplement 1*, panel G.

**Figure supplement 1—source data 2.** Original files for western blot analysis displayed in *Figure 5—figure supplement 1*, panel G.

Next, we performed immunofluorescent staining to study the synaptic distribution of NAT10 proteins in cultured hippocampal neurons with chemical LTP (cLTP) stimulation (*Figure 5—figure supplement 1*) – a cellular model of learning and memory (*Shahi and Baudry, 1993*; *Lu et al., 2001*). In control neurons treated with vehicle, NAT10 was mainly expressed in the nucleus and cytoplasm (*Figure 5—figure supplement 1*), which is consistent with the results of western blots (*Figure 5I, J*). However, 60 min after cLTP stimulation, the co-localization of NAT10 with PSD95, a protein marker for PSD fraction, was significantly increased, compared with control neurons (*Figure 5—figure supplement 1*). These results corroborate the in vivo findings to demonstrate the increase of NAT10 proteins at synapses by neural activities.

## Induction of ac4C modification of *Arc* by cLTP in hippocampal slices

Given our observation that cLTP stimulation promotes the redistribution of NAT10 protein to the PSD, we further examine the effects of cLTP stimulation on the ac4C levels of *Arc*, a typical MISA mRNA (*Figure 3*). The local translation of *Arc* at synapses is known to be activated by neural activities and important for LTP consolidation (*Yin et al., 2002*). We employed acRIP-qPCR to analyze the ac4C levels of *Arc* mRNA in mouse hippocampal slices after cLTP stimulation. The results indicated that ac4C modifications of *Arc* mRNA in the synaptosomes were significantly increased (after being normalized by transcription levels) 1 hr after cLTP stimulation compared with controls (*Figure 5—figure supplement 1*), which is accompanied by the increase of *Arc* proteins in the synaptosomes after cLTP stimulation (*Figure 5—figure supplement 1*). Together with the in vivo findings, these data suggest that neural activities can upregulate ac4C modification of *Arc* mRNA at synapses.

## NAT10-dependent ac4C polyA RNA in mouse hippocampus

To demonstrate the NAT10-dependent ac4C modification of poly A RNA, we used the Cre/Loxp strategy (*Tsien et al., 1996*) to downregulate *Nat10* expression in adult mouse hippocampus through injection of AAV-EGFP-P2A-Cre into the hippocampus of *Nat10*^flox/flox mice. Given that *Nat10* is mainly expressed in the mature neurons (*Figure 4*), the expression of EGFP-2A-Cre in the AAV was driven by the human synapsin 1 (Syn1) promoter, a neuron-specific promoter (*Hedegaard et al., 2013*). The *Nat10*^flox/flox mice injected with AAV-EGFP and AAV-EGFP-P2A-Cre were named control and *Nat10* conditional knockout (cKO) mice hereafter. Both the control and *Nat10* cKO mice received stereotaxic AAV injection into the dorsal hippocampus, and the HOM of dorsal hippocampus was collected for biochemical assay 3 weeks after AAV injection (*Figure 6—figure supplement 1*).

The fluorescent protein EGFP was expressed mainly in the CA1-3 regions and DG of mouse hippocampus 3 weeks after AAV injection (*Figure 6—figure supplement 1*), indicating the successful infection of AAV. The protein levels of NAT10 were significantly reduced in the hippocampus of *Nat10* cKO mice, compared with controls (*Figure 6—figure supplement 1*), verifying the effective knockdown of NAT10 by the Cre/Loxp strategy. NAT10 was previously described as a protein acetyltransferase against α-tubulin and histones (*Lv et al., 2003*; *Shen et al., 2009*). However, immunoblotting with acetylation-specific antibodies showed no significant change of protein acetylation in *Nat10* cKO hippocampus, compared to controls (*Figure 6—figure supplement 1*). These results suggest the redundancy of acetyltransferases for α-tubulin and histones in mouse hippocampus. In contrast, the ac4C modification of polyA RNA was significantly decreased in the hippocampus of *Nat10* cKO mice, compared with controls (*Figure 6—figure supplement 1*). As

a control experiment, downregulation of NAT10 did not affect the m6A modification of polyA RNA (*Figure 6—figure supplement 1*). The LC–MS/MS analysis of polyA RNA indicated that the stoichiometry levels of ac4C rather than m6A were significantly reduced in the hippocampus of *Nat10* cKO mice, compared with controls (*Figure 6—figure supplement 1*). The partial reduction of ac4C in the *Nat10* cKO mice might be due to the limited infection areas of Cre-AAV, or the *Nat10* expression in non-neuronal cells, or the presence of other unknown ac4C writers. Nonetheless, these results demonstrate that NAT10 specifically regulates ac4C rather than m6A modification of polyA RNA in mouse hippocampus.

## NAT10-dependent ac4C mapping of SYN mRNA in memory mice

Given that memory increased ac4C mRNA specifically in the SYN, we next aim to identify NAT10-dependent ac4C modification of synaptic mRNAs. To this end, we performed low-input acRIP-seq for SYN RNA purified from the hippocampus of control and *Nat10* cKO mice at day 6 after memory (*Figure 6—figure supplement 2*). Due to the limited yield of SYN RNA, we purified SYN RNA from the hippocampus of 10 mice and combined them into one sample. Two biological samples from control and *Nat10* cKO mice at day 6 were subjected to low-input acRIP-seq to demonstrate the NAT10-dependent ac4C modification of SYN mRNA. Two libraries (acRIP and input) were sequenced for each biological sample (*Supplementary file 1C*).

We identified 7097 and 1933 ac4C peaks from the SYN of control and *Nat10* cKO mice, respectively (*Figure 6—figure supplement 2*). These results suggest a reduction of the ac4C peak number by knockout of *Nat10* in mature neurons of adult mouse hippocampus. Out of 7097 ac4C peaks identified in control mice, 6331 peaks were abolished and 408 peaks were downregulated in *Nat10* cKO mice, and these ac4C peaks were named <u>NA</u>T10-dependent <u>s</u>ynaptic <u>ac</u>4C (NASA) peaks (*Figure 6—figure supplement 2*) which were mainly localized in the CDS region of mRNA (*Figure 6—figure supplement 2*).

The *Nat10* cKO mice also showed a reduction of ac4C mRNA number in the SYN, compared with controls (*Figure 6—figure supplement 2*). In contrast, the number of mRNA in the SYN was similar between control and *Nat10* cKO mice (*Figure 6—figure supplement 2*). Likewise, the levels of ac4C mRNA rather than total mRNA were significantly reduced in the SYN of *Nat10* cKO mice, compared with controls, evidenced by the leftward-shifted cumulative distribution curve of FE of ac4C mRNA in the *Nat10* cKO mice (*Figure 6A–C*). Out of 3778 ac4C-modified mRNAs identified in the SYN of control mice, 2507 were classified as diminished, indicating that their ac4C enrichment fell below the detection threshold or became undetectable in *Nat10* cKO mice (FE <1 or peak lost). An additional 839 mRNAs were downregulated, defined as showing significantly reduced but still detectable ac4C levels in cKO mice compared to controls. Collectively, these transcripts were referred to as NASA mRNAs (*Figure 6—figure supplement 2*; *Supplementary file 1E*). Analysis of the overlap between MISA and NASA indicated that 717 out of 1237 MISA mRNAs were regulated by NAT10 (*Figure 6D*).

To further investigate the molecular basis underlying the differential regulation of MISA mRNAs by NAT10, we performed sequence motif analysis on the two distinct gene populations – 717 NAT10-dependent MISA mRNAs (present in both MISA and NASA populations) and 520 NAT10-independent MISA mRNAs (present only in MISA but absent in NASA) (*Figure 6D*). Remarkably, these two gene sets exhibited distinct ac4C consensus motifs (*Figure 6D*). The NAT10-dependent MISA mRNAs displayed the canonical AGCAGCTG motif that was consistently observed throughout our study. In contrast, the NAT10-independent MISA mRNAs showed a fundamentally different motif pattern, characterized by RGGGCACTAACY sequence. The complete divergence in motif sequences between these two populations suggests that the NAT10-independent MISA mRNAs may be regulated by as yet unidentified ac4C writers.

We next analyzed the cellular expression pattern of NASA mRNAs through the single-cell RNA-seq atlas of mouse hippocampus (dropviz.org) (*Saunders et al., 2018*). The results indicated that the expression of NASA mRNAs was mostly enriched in excitatory neurons (*Figure 6—figure supplement 2*), which is in line with the cellular expression pattern of NAT10 in mouse hippocampus (*Figure 4*). In addition, about 13% of NASA mRNAs were relatively enriched in glia cells (*Figure 6—figure supplement 2*), which is consistent with the notion that processes of glia cells contribute to formation of the tripartite synapses (*Perea et al., 2009*; *Eroglu and Barres, 2010*).

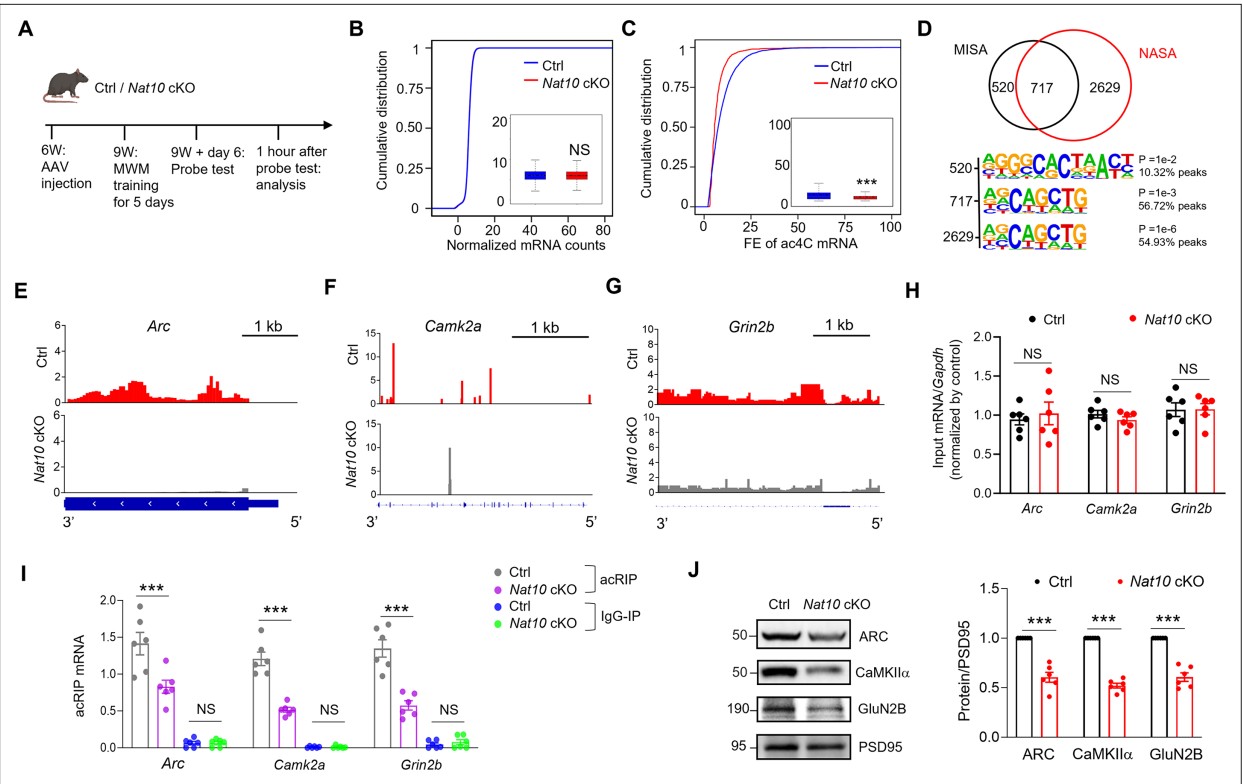

**Figure 6.** NAT10-dependent ac4C modification of memory-related mRNAs in the SYN of memory mice. (**A**) Experimental design. Three weeks after AAV injection, the control (Ctrl) and *Nat10* conditional knockout (cKO) mice were subjected to Morris water maze (MWM) training and probe test, and then the SYN fractions were purified from hippocampal tissues for low-input acRIP-seq. (**B**) The total mRNA levels were not significantly altered in the SYN of *Nat10* cKO mice, compared to controls. Shown are cumulative distribution curves of normalized mRNA counts for SYN in control and *Nat10* cKO mice at day 6 after MWM training. The normalized mRNA counts were quantified in the quartile boxplots, where the solid line in the box is the median and the dashed line is the mean, where the maximum and minimum values are identified. NS, not significant, Kolmogorov–Smirnov test. (**C**) Acetylation of mRNA in the SYN of *Nat10* cKO mice was significantly reduced, compared with controls. Shown are cumulative distribution curves of fold enrichment (FE) of ac4C mRNA purified from the SYN of control and *Nat10* cKO mice at day 6 after MWM training. The FE of ac4C mRNA was quantified in the quartile boxplots, where the solid line in the box is the median and the dashed line is the mean, where the maximum and minimum values are identified. ***adj-p = 1.20076e−12, Kolmogorov–Smirnov test. (**D**) Venn diagrams show overlapping between MISA mRNA and NASA mRNA. Distinct consensus motifs distinguish NAT10-dependent and NAT10-independent MISA mRNAs. Sequence motif analysis reveals fundamentally different consensus sequence patterns between the two MISA mRNA populations identified in the Venn diagram. Reduced ac4C modification of *Arc* (**E**), *Camk2a* (**F**), and *Grin2b* (**G**) mRNA in the SYN of *Nat10* cKO mice at day 6, compared with controls. Shown are the IGV maps of the ac4C peaks from the 3′ UTR to the 5′ UTR of transcripts. The ac4C peaks were represented by acRIP/input (see details in the Method). (**H**) The mRNA levels of *Arc*, *Camk2a*, and *Grin2b* from the input of SYN were similar between control and *Nat10* cKO mice. The RNA purified from the SYN was subjected to the RT-qPCR analysis. NS, not significant, two-tailed *t*-test, n = 6 biological replicates. The levels of mRNA were normalized by that of *Gapdh* in each sample, and then the ratio of mRNA to *Gapdh* was normalized by controls. (**I**) The ac4C modification of *Arc*, *Camk2a*, and *Grin2b* mRNA was significantly reduced in the SYN of *Nat10* cKO mice at day 6, compared with controls. The RNA purified from the acRIP and IgG-IP products of SYN was subjected to the RT-qPCR analysis. ***p < 0.001, two-way-ANOVA followed by Tukey's multiple comparison test, n = 6 biological replicates. Data were normalized by input. (**J**) The protein levels of ARC, CaMKIIα, and GluN2B were significantly decreased in the SYN of *Nat10* cKO mice at day 6, compared with controls. Left, representative western blot images; right, quantification data. ***p < 0.001, two-tailed *t*-test, n = 6 biological replicates. The protein levels were normalized by that of PSD95 in each sample, and then the ratio of protein to PSD95 was normalized by controls. Quantification data are expressed as mean ± SEM.

The online version of this article includes the following source data and figure supplement(s) for figure 6:

**Source data 1.** PDF file containing original western blots for *Figure 6J*.

**Source data 2.** Original files for western blot analysis displayed in *Figure 6J*.

**Figure supplement 1.** NAT10-dependent ac4C modification of polyA RNA in mouse hippocampus.

**Figure supplement 1—source data 1.** PDF file containing original membranes corresponding to *Figure 6—figure supplement 1*, panel C.

**Figure supplement 1—source data 2.** Original files for western blot analysis displayed in *Figure 6—figure supplement 1*, panel C.

**Figure supplement 2.** NAT10-dependent ac4C mapping in the SYN of memory mice.

## Regulation of ac4C and protein levels of memory-related mRNAs by NAT10

*Arc*, a representative MISA mRNA (*Figure 3*), is critical for memory (*Bramham et al., 2008*). The mRNA of $Ca^{2+}$/calmodulin-dependent protein kinase 2α (*Camk2a*) and *Grin2b* encode the protein kinase CaMKIIα and the GluN2B subunit of NMDA receptor, respectively. CaMKIIα and GluN2B are synapse-enriched proteins and play important roles in LTP as well as memory (*Lisman et al., 2012*; *Tang et al., 1999*). Recent studies demonstrate that CaMK2α interaction with GluN2B is critical for the induction and maintenance of LTP (*Tullis et al., 2023*; *Chen et al., 2024*). Analysis of the low-input acRIP-seq data indicated the reduction of ac4C peaks of *Arc*, *Camk2a,* and *Grin2b* mRNAs in the SYN of *Nat10* cKO mice, compared with controls (*Figure 6E–G*). The genome browser tracks for the ac4C of all NASA mRNAs were accessible at the ac4C website (http://ac4catlas.com/) to show reduced or diminished ac4C peaks in the SYN of *Nat10* cKO mice, compared with controls. Despite the presence of a common AGCAGCTG motif in both groups, *Nat10* deletion disrupted the local CXX sequence context, reflecting a shift in ac4C deposition specificity upon loss of *Nat10* enzymatic activity (*Figure 6—figure supplement 2*). Deconvolution with CIBERSORT identified neurons as the top enriched cell type for both NAT10-dependent and -independent MISA mRNAs. The neuronal enrichment did not differ between the two sets, which argues against differences in cellular origin (*Figure 6—figure supplement 2*). Moreover, BP enrichment of the 717 shared ac4C-modified mRNAs revealed significant enrichment in BPs such as synapse organization and dendrite development, indicating a potential role of ac4C modification in regulating synaptic structure and neuronal connectivity (*Figure 6—figure supplement 2*).

Next, we performed RT-qPCR analysis for input and acRIP RNAs purified from the SYN of control and *Nat10* cKO mice at day 6 after memory to verify the NAT10-dependent ac4C modification of *Arc*, *Camk2a*, and *Grin2b* mRNAs. The input mRNA levels of *Arc*, *Camk2a*, and *Grin2b* in the SYN were comparable between control and *Nat10* cKO mice (*Figure 6H*). In contrast, the acRIP mRNA levels of *Arc*, *Camk2a*, and *Grin2b* in the SYN were significantly reduced in *Nat10* cKO mice, compared with controls (*Figure 6I*). Given that ac4C modification of mRNA has been shown to enhance translation efficiency (*Arango et al., 2018*; *Arango et al., 2022*), we sought to determine whether NAT10 influences the protein levels of ARC, CaMKIIα, and GluN2B in the SYN by western blot analysis. The protein levels of ARC, CaMKIIα, and GluN2B were significantly decreased in the SYN at day 6 from *Nat10* cKO mice, compared with controls (*Figure 6J*). Altogether, these results demonstrate that NAT10 regulates ac4C and protein levels of *Arc*, *Camk2a*, and *Grin2b* in the SYN of mouse hippocampus at day 6.

## Regulation of LTP as well as memory by NAT10 in mouse hippocampus

In the following study, we investigate whether NAT10 in mouse hippocampus is important for synaptic plasticity. The body and brain weight were similar between control and *Nat10* cKO mice (*Figure 7—figure supplement 1*). The number and intrinsic excitability of CA1 pyramidal neurons were also comparable between control and *Nat10* cKO mice (*Figure 7—figure supplement 1*). To address whether NAT10 regulates LTP, we performed electrophysiology recording of fEPSP (field excitatory postsynaptic potential) at the SC-CA1 synapses before and after high-frequency stimulation (HFS) (*Figure 7A*). The induction of LTP was comparable between control and *Nat10* cKO mice (*Figure 7B–E*). However, the maintenance of LTP, especially the late phase of LTP, was severely impaired in *Nat10* cKO mice, compared with controls (*Figure 7B, C, F*).

To further study whether *Nat10* is important for synaptic plasticity after memory, we performed fEPSP recording to investigate glutamatergic transmission at the SC-CA1 synapses in the control and *Nat10* cKO mice at day 6 (*Figure 7G*). Consistent with the previous report (*Su et al., 2023*), the glutamatergic transmission at the SC-CA1 synapses was significantly enhanced at day 6 in memory mice, compared with mice without memory (*Figure 7H, I*). The glutamatergic transmission at the SC-CA1 synapses was similar between control and *Nat10* cKO mice without memory, which indicates that *Nat10* may be dispensable for the basal glutamatergic transmission at the SC-CA1 synapses. However, the memory-induced enhancement of glutamatergic transmission at the SC-CA1 synapses was significantly impaired in *Nat10* cKO mice, compared with controls (*Figure 7H, I*). Altogether, these results indicate that NAT10 is important for memory-related synaptic plasticity in mouse hippocampus.

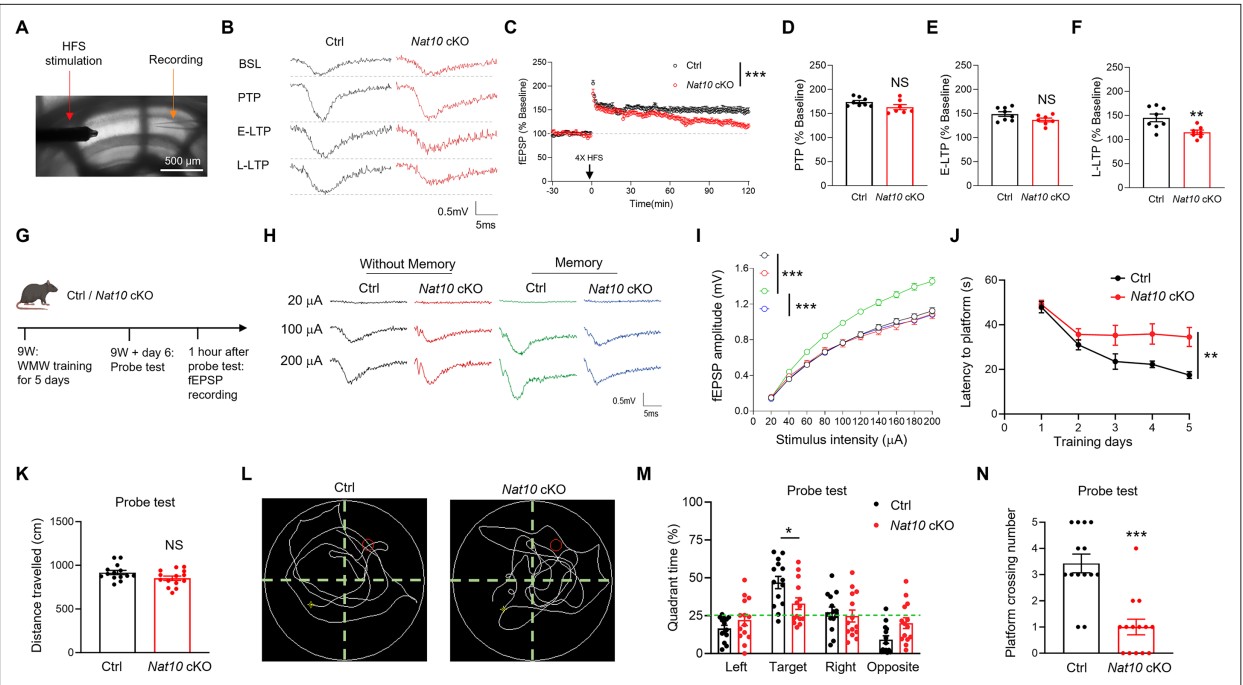

**Figure 7.** Impaired LTP as well as memory in the *Nat10* cKO mice. (**A**) A representative image showing recording LTP at the SC-CA1 synapses from *Nat10* cKO mice using the protocol of high-frequency stimulation (HFS). Scale bar, 500 μm. (**B**) The representative fEPSP traces from baseline (BSL), post-tetanus potentiation (PTP), early phase of LTP (E-LTP, 55–60 min after HFS), and late phase of LTP (L-LTP, 115–120 min after HFS). (**C**) Impaired maintenance of L-LTP at the SC-CA1 synapses in *Nat10* cKO mice, compared with controls. Normalized fEPSP amplitudes were plotted every 1 min for hippocampal slices from control and *Nat10* cKO mice. ***p < 0.001, two-way ANOVA, n = 8 slices from 4 mice for each group. (**D**) Normal LTP induction at the SC-CA1 synapses in *Nat10* cKO mice, compared to controls. The PTP in panel C was quantified. NS, not significant, two-tailed *t*-test. n = 8 slices from 4 mice for each group. (**E**) Unaltered E-LTP at the SC-CA1 synapses in *Nat10* cKO mice, compared to controls. The E-LTP in panel C was quantified. NS, not significant, two-tailed *t*-test. n = 8 slices from 4 mice for each group. (**F**) Impaired L-LTP at the SC-CA1 synapses in *Nat10* cKO mice, compared with controls. The L-LTP in panel C was quantified. **p = 0.008, two-tailed *t*-test. n = 8 slices from 4 mice for each group. (**G**) Experimental design. The control and *Nat10* cKO mice were subjected to fEPSP recording at the SC-CA1 synapses after training and probe test in the Morris water maze (MWM). (**H, I**) Memory-induced enhancement of glutamatergic transmission at SC-CA1 synapses was significantly impaired in *Nat10* cKO mice, compared with controls. (**H**) Representative fEPSP traces from control and *Nat10* cKO mice. (**I**) Quantification of the fEPSP amplitudes in panel H. ***Genotype $F_{(3, 680)}$ = 128.2, p < 0.001, two-way ANOVA followed by Sidak's multiple comparison test, n = 18 slices from 5 mice each group. (**J**) Impaired memory process in *Nat10* cKO mice, compared with controls. Shown are the memory curves of control and *Nat10* cKO mice in the MWM. n = 14 mice per group, **Genotype $F_{(1, 130)}$ = 24.08, **p = 0.0014, two-way ANOVA followed by Sidak's multiple comparison test. (**K**) Similar distance traveled during probe tests between control and *Nat10* cKO mice. NS, not significant, two-tailed t-test, n=14 mice per group. (**L**) Representative swimming traces of control and *Nat10* cKO mice during probe tests. (**M**) Time spent in the target quadrant was significantly reduced in *Nat10* cKO groups during probe tests, compared with controls. *p = 0.0198, two-way ANOVA followed by Sidak's multiple comparison with post hoc *t*-test, n = 14 mice per group. (**N**) Number of platform crossings was significantly reduced in *Nat10* cKO mice during probe tests, compared with controls. ***p < 0.001, two-tailed *t*-test, n = 14 mice per group. Quantification data are expressed as mean ± SEM.

The online version of this article includes the following figure supplement(s) for figure 7:

**Figure supplement 1.** Downregulation of NAT10 in mouse hippocampus does not affect body or brain weight, neuronal number, or intrinsic excitability.

To investigate whether NAT10 is critical for memory, both control and *Nat10* cKO mice were subjected to training in the MWM for 5 days, and the probe test was performed at day 6. Both groups exhibited an overall decline in the latency to reach the platform at the early stage (days 1–2) of training (*Figure 7J*). However, the *Nat10* cKO mice failed to improve performance during the late training stage (days 3–5) compared with controls (*Figure 7J*). During the probe tests, the control and *Nat10* cKO mice showed similar distance traveled (*Figure 7K, L*). However, the time spent in the target area and the number of platform crossings were significantly reduced in the *Nat10* cKO mice, compared with controls (*Figure 7L–N*). These results demonstrate that NAT10 in mouse hippocampus is important for spatial memory.

## Discussion

In this study, we demonstrated the dynamic regulation of synaptic mRNA acetylation by memory in mouse hippocampus. We identified 1237 MISA mRNAs that were increased in the SYN of mouse hippocampus after memory but returned to normal levels after memories were naturally forgotten. Interestingly, the protein levels of NAT10 were also increased in the SYN after memory but returned to control levels after forgetting. We further showed NAT10-dependent ac4C mapping of SYN mRNAs through the acRIP-seq and verified the NAT10-dependent ac4C modification of *Arc*, *Camk2a*, and *Grin2b* mRNAs. The maintenance of LTP at the SC-CA1 synapses as well as the behaviors of memory were severely impaired in the *Nat10* cKO mice. In support of the importance of ac4C in memory, our recent study showed that ac4C modification of several MISA mRNAs including *Arc*, *Grin1*, and *Psen1* was significantly reduced in the hippocampus of 5× FAD mouse, an animal model of Alzheimer's disease that was characterized by deficient synaptic plasticity and memory (*Ji et al., 2025*). Altogether, these results demonstrate the importance of NAT10-mediated mRNA acetylation in memory.

The dynamic regulation of protein acetylation in the SYN by memory has been shown in our previous studies (*Zhang et al., 2021a*; *Zhang et al., 2021b*; *Zhang et al., 2021c*; *Zhang et al., 2024*). In contrast to the well-characterized acetyltransferases and deacetylases for protein acetylation (*Drazic et al., 2016*), the writer and eraser of mRNA acetylation are less well understood. The NAT10 acetyltransferase is the only known ac4C writer in mammalian cells (*Hu et al., 2024*; *Cui et al., 2023*). Recently, the deacetylase SIRT7 has been shown to erase the ac4C modification of rRNA and small nucleolar RNA (snoRNA) (*Xu et al., 2021*; *Kudrin et al., 2021*). However, the mechanisms underlying the deacetylation of mRNA need further exploration.

A pivotal and unexpected finding of our study fundamentally expands the regulatory landscape of ac4C in neurons. We discovered that a substantial fraction of MISA mRNAs (520 out of 1237, approximately 42%) is acetylated in a NAT10-independent manner. This is not a minor detail, but rather points to the existence of a parallel regulatory system for synaptic mRNA acetylation that is co-activated during memory formation. Strikingly, bioinformatic analysis revealed that these NAT10-independent targets possess a consensus motif (RGGGCACTAACY) completely distinct from the canonical motif associated with NAT10. This strongly suggests the involvement of at least one other, as-yet-unidentified, ac4C writer enzyme with a different sequence specificity. This possibility is supported by the reports of NAT10-independent acetylation in other biological contexts (*Svobodová Kovaříková et al., 2023*; *Schiffers and Oberdoerffer, 2024*; *Liu et al., 2023b*). The existence of this robust parallel pathway may also explain why the *Nat10* cKO phenotype, while significant, is not absolute. This discovery transforms the narrative from a single-enzyme mechanism to a more complex, multi-faceted regulatory network and opens up critical new questions for future research regarding the identity of this novel writer and the functional specialization of these parallel ac4C pathways in synaptic plasticity.

How the ac4C modification of mRNA is regulated by neural activities remains largely unknown. Our previous studies showed that overnight starvation downregulated the ac4C modification of tyrosine hydroxylase (TH) mRNA in the sympathetic nerve terminus that innervates the olfactory mucosa in mice, which reduced TH protein levels and dopamine synthesis in the mouse olfactory mucosa after overnight starvation (*Zhou et al., 2022*). In this study, we demonstrate that memory regulates the dynamic change of ac4C modification of mRNA at synapses of mouse hippocampus. Altogether, these studies provide evidence that neural activities regulate acetylation of mRNA in a spatiotemporally specific manner.

NAT10 proteins were increased in the SYN but reduced in the cytoplasm of mouse hippocampus at day 6 in the MWM. These results suggest a relocation of NAT10 from the cytoplasm to synapses after memory, which was corroborated by the immunostaining study of NAT10 distribution in cultured mouse hippocampal neurons after cLTP stimulation. Previous proteomic studies showed that NAT10 was detected in the interactome with several motor proteins such as kinesin proteins (*Rouillard et al., 2016*), which raised the possibility that NAT10 might be transported by the motor proteins in hippocampal neurons during memory. NAT10 was highly expressed in the nucleus of hippocampal neurons, but protein levels of nuclear NAT10 were not altered at day 6 in the MWM. NAT10 has been shown to be important for regulating nuclear functions such as nucleolar assembly (*Shen et al., 2009*) and nuclear architecture (*Larrieu et al., 2014*). The function of NAT10 in the nucleus of neurons waits for future investigations.

The subcellular change of NAT10 proteins at day 6 apparently may not be caused by stress during the training in the MWM. The mice were well handled before and during the training, and thus the stress should be minor, if any. The control mice were exposed to the training using a visible platform on day 1. The stress levels in the control mice may vary with the memory mice. If mild stress during the training in the MWM could cause the relocation of NAT10 to the SYN, the protein levels of NAT10 in the SYN would be increased at day 1. However, the protein levels of NAT10 in the SYN were similar at day 1 between control and memory mice. In addition, chronic mild stress has been shown to increase the total protein levels of NAT10 in mouse hippocampus (*Guo et al., 2022*). In contrast, the total protein levels of NAT10 in mouse hippocampus did not alter at day 6 in the MWM. These results suggest that the increased NAT10 proteins in the SYN at day 6 in the MWM may be related to memory rather than stress.

The stoichiometry levels of ac4C in polyA RNA have been reported to be ~0.2% in HeLa cells (*Arango et al., 2018*), ~0.1% in human embryonic stem cells (*Hu et al., 2024*), ~0.2% in liver (*Hao et al., 2022*), and ~0.3% in heart (*Shi et al., 2023*). Here we showed that ~0.4% of cytidine was acetylated in mouse hippocampus. Although the total protein levels of NAT10 have been reported to be relatively lower in the whole brain tissue compared with reproductive tissues such as testis and ovary (*Jiang et al., 2023*), NAT10 was shown to be highly expressed in hippocampal neurons (*Gao et al., 2024*). Future studies are warranted to investigate how the acetyltransferase activity of NAT10 toward mRNA was regulated in different brain regions and different tissues.

The basal glutamatergic transmission at the SC-CA1 synapses was unaltered in the *Arc* knockout mice (*Plath et al., 2006*), which is consistent with the phenotypes in the *Nat10* cKO mice. On the other hand, ARC has been shown to accelerate AMPA receptor endocytosis (*Chowdhury et al., 2006*), and thus the induction of LTP at the SC-CA1 synapses was enhanced in the *Arc* knockout mice (*Plath et al., 2006*). Although ARC protein levels were decreased in the SYN of *Nat10* cKO mice, the induction of LTP at the SC-CA1 synapses was not increased in the *Nat10* cKO mice, which might be due to the reduction of other LTP-promoting proteins such as CaMKIIα and GluN2B in the SYN of *Nat10* cKO mice.

In line with the phenotypes of *Nat10* cKO mice presented in this study, the *Arc* knockout mice showed deficits in the late phase of LTP at SC-CA1 synapses and impairment of memory consolidation in the MWM (*Plath et al., 2006*). Consistent with the function of *Arc* in memory consolidation, the acetylation of *Arc* mRNA was increased in the SYN at day 6 rather than day 1 following MWM training, a period during which memories were consolidated. Furthermore, our findings from cLTP in mouse hippocampal slices suggest that this acetylation process can be induced rapidly. We observed a significant increase in the ac4C levels of *Arc* mRNA in mouse hippocampal slices 1 hr after cLTP stimulation, a timescale consistent with the consolidation phase of LTP, for which sustained local synthesis of *Arc* is known to be critical (*Bramham et al., 2008*; *Messaoudi et al., 2007*). In sum, the data presented here reveal a novel mechanism underlying how neural activity regulates local protein synthesis at synapses using *Arc* as an example.

Although our current study primarily focuses on ac4C modifications of mRNA, we cannot exclude the possibility that *Nat10* deficiency also affects post-translational regulatory pathways. Moreover, the function of NAT10 in regulating tRNA or rRNA might also contribute to memory. New techniques for overexpression and erasure of site-specifically acetylated mRNA need to be developed to precisely study the function of mRNA acetylation. Nonetheless, the results presented here provide evidence that NAT10-mediated mRNA acetylation may contribute to the protein synthesis at synapses during memory in mouse hippocampus.

## Methods

### Animals

Male C57BL/6 mice were used throughout this study. For stereotaxic injection of AAV, 6-week-old mice were used. For behavioral analysis, 9- to 10-week-old mice were used. For ac4C dot-blots, LC–MS/MS, acRIP-seq, mice aged between 9 and 10 weeks were used. For immunofluorescence, western blots, and electrophysiology, 10-week-old mice were used. Animals were housed in rooms at 23°C and 50% humidity in a 12-hr light/dark cycle and with food and water available ad libitum. Animal experimental procedures were approved by the Institutional Animal Care and Use Committee of East

China Normal University. The Ai14 Cre-dependent tdTomato reporter mouse line (stock no. 007914) was obtained and used for generating reporter mice.

## Generation of Nat10^{Cre/+} knockin mice

The mouse *Nat10* gene has 29 exons and the translational start codon (ATG) is localized in exon 2. We employed CRISPR/Cas9 technology in conjunction with homologous recombination to insert a Cre-WPRE-polyA expression cassette at the translational start codon of the *Nat10* gene in C57BL/6J mice. The process commenced with the synthesis of Cas9 mRNA and a guide RNA (gRNA) targeting the start codon of *Nat10*, with the gRNA sequence being 5'-catcatgaatcggaagaaggtgg-3'. A donor vector was then constructed using In-Fusion cloning, containing 2.6 kb 5' and 3' homology arms flanking the Cre-WPRE-pA cassette, designed for Cre recombinase expression, enhanced by the WPRE element and concluded with a polyadenylation signal. The cocktail of Cas9 mRNA, gRNA, and the donor vector was microinjected into fertilized C57BL/6J mouse embryos, leading to the birth of F0 generation mice. These mice were screened for the precise genomic integration of the cassette through PCR amplification and sequencing. Successful modifications were observed in F0 mice, which were subsequently bred with C57BL/6J mice to secure F1 progeny harboring the modification.

The resulting offspring were genotyped using PCR, with primers designed to detect the Cre within the *Nat10* gene. The primers for genotyping wt *Nat10* allele are as follows: forward: 5'tgtctttcct gtggtggtgt3'; reverse: 5'tgcgcactgaaacctacaa3'. The primers for genotyping *Nat10*-cre allele are as follows: forward: 5'gaaatcatgcaggctggtgg3'; reverse: 5'aaagtcccggaaaggagctg3'. The PCR products for wt *Nat10* and *Nat10*-Cre alleles were 392 and 393 bp, respectively.

## Generation of Nat10^{flox/flox} mice

The generation of *Nat10*^{flox/flox} mice involved targeted modification of exon 4 within the *Nat10* gene, utilizing CRISPR/Cas9-mediated gene editing. Two gRNAs were specifically designed to target and flank exon 4, enabling a precise floxed modification. The sequences for these gRNAs were as follows: gRNA1: 5'-tcagcagaccactttaaagatgg-3' and gRNA2: 5'-tgcctgacatagtaaggtctggg-3'. Both Cas9 mRNA and gRNAs were produced through in vitro transcription techniques. A donor vector for homologous recombination was constructed using In-Fusion cloning. This vector included a 3.0-kb 5' homology arm, a 0.7-kb loxp-flanked region (flox), and a 3.0-kb 3' homology arm to ensure accurate genomic insertion at the *Nat10* locus. The mixture of Cas9 mRNA and the two gRNAs was injected into the cytoplasm of one-cell stage embryos. These embryos were subsequently implanted into pseudopregnant females and carried to term. The resulting offspring were genotyped using PCR, with primers designed to detect the floxed exon 4 of *Nat10* gene. The primers for genotyping the wt and floxed *Nat10* allele are as follows: forward: 5'tgccgaggggatgtactcat3'; reverse: 5'agaaccccacgaactgtcct3'. The PCR products for wt *Nat10* and floxed *Nat10* allele were 302 and 356 bp, respectively.

## Morris water maze

The MWM, a widely recognized method for assessing spatial memory (*Vorhees and Williams, 2006*), was implemented with the following protocol: mice were acclimated to the testing environment over a period of 3 days through 10-min daily handling sessions. To aid navigation, visual cues were strategically positioned around the perimeter of the maze. To ensure consistency and minimize the impact of external variables, all experimental sessions were conducted simultaneously each day. During the core phase of the experiment, spanning 5 days, mice were subjected to four trials daily. In each trial, they were released from different quadrants into the maze, with the escape platform being fixed in the first quadrant. A trial was deemed complete when a mouse successfully located the platform within a 60-s timeframe. If unsuccessful, the mouse was gently guided to the platform at the 60-s mark. The mice were handled for 10 min each day following the completion of the four trials.

On day 6, a probe test was conducted to assess memory retention: the platform was removed, and mice were released from the third quadrant. The number of crossings over the previous platform location during the 60-s trial was recorded. To evaluate visual and motor function independent of spatial memory, a visible platform test was conducted on a separate cohort of control mice. The platform was made visible by placing a flag above the surface, and its location was changed between trials to prevent spatial memory. Mice underwent four trials in 1 day, with performance measured by latency to

reach the visible platform. This design was chosen to match handling and navigation exposure, while minimizing spatial memory.

On day 20, a second probe test was conducted using the same conditions as on day 6, to assess long-term memory retention after a 2-week interval in the home cage. Control mice underwent both the visible platform test and were subjected to identical handling procedures and probe test conditions on days 6 and 20. This design ensured that differences observed in probe test performance reflected spatial memory rather than sensory or motor deficits. Mice designated for molecular or histological analysis were euthanized 60 min after the probe test began. This time point was selected based on established literature indicating that memory retrieval-related molecular signatures, such as the activation of immediate early genes and RNA modifications, occur at this time (*Guzowski et al., 2001*).

## Chemical and reagents

All chemicals and reagents utilized in this study were of analytical grade. RNA was extracted using TRIzol Reagent (15596-026, Invitrogen, Carlsbad, CA, USA) following the manufacturer's protocol. Chloroform and other general laboratory chemicals were obtained from Merck (109634, Darmstadt, Germany). The ACQUITY HSS T3 column (2.1 × 100 mm, 1.8 μm) for LC–MS/MS analysis was purchased from Waters Corporation (186008499, Milford, MA, USA). Phosphodiesterase was obtained from Sigma (P3243-1VL, St. Louis, MO, USA). Nuclease P1 and alkaline phosphatase, used for RNA digestion, were obtained from Takara (2410A and 2250A). Hydroxylamine was from Aladdin Bio-Chem Technology Co, Ltd (H164487). RNase inhibitor, which is critical for preserving RNA integrity during manipulations, was purchased from New England Biolabs (M0314S, NEB, CA, USA). Methylene blue, which is used for nucleic acid visualization, was obtained from Abcam (M9140). Poly-D-lysine was purchased from Sigma (Cat. No. P1149). Tamoxifen was purchased from Sigma (Cat T5648).

## Subcellular fractionation

The procedure of subcellular fractionation was adapted from the methodologies in our previous studies (*Zhang et al., 2021a*; *Zhang et al., 2021b*). Mouse brain tissues were processed starting with homogenization in Buffer A, which consists of 0.32 M sucrose, 1 mM $MgCl_2$, 1 mM PMSF, and a protease inhibitor cocktail to ensure comprehensive cell lysis while preserving protein integrity. The resultant homogenates were then filtered to remove cell debris and subsequently subjected to centrifugation at 500 × $g$ for 5 min using a fixed-angle rotor. This step separates the homogenate into the pellet (P1) and supernatant (S1) fractions. The P1 fraction, containing primarily nuclei and unbroken cells, was further purified by resuspension in Buffer B (10 mM KCl, 1.5 mM $MgCl_2$, and 10 mM Tris-HCl at pH 7.4) followed by a repeat centrifugation under the same conditions to remove any remaining debris. The final pellet was resuspended in Buffer C (20 mM HEPES, pH 7.9, 25% glycerol, 1.5 mM $MgCl_2$, 1.4 M KCl, 0.2 mM EDTA, 0.2 mM PMSF, and 0.5 mM DTT), which is designed for nuclear protein extraction and incubated with agitation at 4°C for 30 min to facilitate the release of nuclear proteins. A final centrifugation at 12,000 × $g$ for 10 min was performed, and the supernatant, containing the nuclear protein fraction, was collected. The initial supernatant (S1), following the removal of P1, was further centrifuged at 10,000× $g$ for 10 min to partition it into the P2 fraction, which included the crude synaptosomes, and the cytosolic S2 fraction.

## Western blot

Homogenates of hippocampus tissue were prepared in the RIPA buffer containing 50 mM Tris-HCl, pH 7.4, 150 mM NaCl, 2 mM EDTA, 1% sodium deoxycholate, 1% SDS, 1 mM PMSF, 50 mM sodium fluoride, 1 mM sodium vanadate, 1 mM DTT, and protease inhibitor cocktails. All the protein samples were boiled at 100°C water bath for 10 min before western blot. Homogenates were resolved on SDS/PAGE and then transferred to nitrocellulose membranes, which were incubated in the TBS buffer (0.1% Tween-20 and 5% milk) for 1 hr at room temperature (RT) followed by incubation of primary antibodies overnight at 4°C. After washing, the membranes were incubated with HRP-conjugated secondary antibody in the same TBS buffer for 2 hr at RT. Immunoreactive bands were visualized by ChemiDocTM XRS + Imaging System (Bio-Rad) using enhanced chemiluminescence (Pierce) and analyzed with Image J (NIH). The following primary antibodies were used: anti-Histone H3 (1:1000, Cell Signaling, 9715), anti-SYN1 (1:1000, Cell Signaling, 2312), anti-α-tubulin (1:10,000, Cell Signaling,

3873), anti-NAT10 (1:1000, Abcam, ab194297), anti-Acetyl-Histone H3 (1:1000, Cell Signaling, 9649), anti-ARC (1:1000, Proteintech, 66550-1-Ig), anti-PSD95 (1:1000, Cell Signaling, 3450), anti-PCAF (1:1000, Cell Signaling, 3378), and anti-GAPDH (1:5000, Abways, ab0037). The following secondary antibodies were used: HRP-conjugated secondary antibody (goat-anti-mouse, G-21040; goat-anti-rabbit, G-21234; 1:2000, Thermo Fisher).

### Immunofluorescence

The mouse brains were fixed in 4% paraformaldehyde at RT for 1 hr and dehydrated in 20% sucrose at 4°C overnight. Then the tissues were placed in an embedding mold and frozen in OCT media at 0°C. The embedded tissues were cut into slices with the thickness of 30 µm using a Leica microtome (CM3050S). The brain slices were permeabilized with 0.3% Triton X-100 and 5% BSA in PBS and incubated with primary antibodies at 4°C overnight. After washing with PBS for three times, samples were incubated with secondary antibodies for 2 hr at RT. Samples were mounted with Vectashield mounting medium (Vector Labs) and images were taken by Leica TCS SP8 confocal microscope. The following primary antibodies were used: rabbit anti-NeuN (1:1000, Abcam, ab177487), rabbit anti-Neurogranin (1:500, Abcam, ab217672), rabbit anti-Cre (1:500, CST, 15036S), mouse anti-GFAP (1:1000, Millipore, MAB360), mouse anti-NeuN (1:1000, Abcam, ab104224). The following secondary antibodies were employed: Rabbit IgG (H+L) Highly Cross-Adsorbed Secondary Antibody (A32732, 1:500, Thermo Fisher). Unbiased stereology TissueFAX Plus ST (Tissue Gnostics, Vienna, Austria) was applied to count the number of *Nat10*-positive neurons. The detailed methods for cell number counting are available on the website here.

### Detection of ac4C and m6A modifications by LC–MS/MS

1 µg of total RNA or polyA RNA was digested by 4 µl of nuclease P1 in 40 µl buffer (10 mM Tris-HCl, pH 7.0, 100 mM NaCl, 2.5 mM ZnCl$_2$) at 37°C for 12 hr, followed by incubation with 1 µl of alkaline phosphatase at 37°C for 2 hr. The RNA solution was then diluted to 100 µl and analyzed by LC–MS/MS. The separation of nucleosides was accomplished by means of an Agilent C18 column, and detection was achieved by AB SCIEX QTRAP 5500. Standard curves were generated to reflect the linear relationship between the ac4C (or C) concentration and their peak area during the LC–MS/MS analysis. A known concentration of ac4C or C that falls into the linear range of the standard curve (i.e, the standard samples) was analyzed alongside the testing samples. The concentration of ac4C or C in the testing samples was calculated through normalization of the peak areas from testing samples by that from standard samples. The stoichiometry levels of ac4C were determined by the ratio of concentration of ac4C to C. The same strategies were used for analysis of the stoichiometry levels of m6A.

### ac4C and m6A dot-blot

The total RNA of mouse hippocampus was extracted with Trizol according to the manufacturer's instructions. The polyA RNA was then purified using the Dynabeads polyA RNA purification kit (61006, Thermo Fisher). The dot-blot was performed according to the previous study (*Zhang et al., 2023b*). We used 0.5 µg of polyA RNA that falls in the linear range for each blot. For each HOM replicate, we purified 0.5 µg of polyA RNA from the hippocampus of one mouse. In contrast, for each SYN replicate, we purified 0.5 µg of polyA RNA from the hippocampus of five mice. In brief, the polyA RNA was denatured at 95°C for 5 min using a heat block, immediately kept on ice for 1 min, loaded onto Amersham Hybond-N+membrane and crosslinked with UV254 for 30 min. The membranes were blocked with 5% nonfat milk for 2 hr at RT and incubated with the rabbit anti-m6A (1:2000, Synaptic System, #202003) or anti-ac4C antibody (1:500, Abcam, ab252215) at 4°C overnight. After washing with 0.05% PBST for three times, the membranes were incubated with the HRP-conjugated secondary antibody (goat-anti-rabbit, G-21234, 1:2000, Thermo Fisher) at RT for 2 hr. The signals were visualized by ChemiDocTM XRS + Imaging System (Bio-Rad) using ECL and analyzed with Image J (NIH). RNA loading was verified by staining the membranes with 0.2% methylene blue. The intensity of each dot was normalized to total RNA.

### Primary neuronal culture

The primary culture of mouse hippocampal neurons was performed according to our previous studies (*Ting et al., 2011*). Hippocampal explants from embryonic day 16 (E16) mouse embryos were

digested with 0.125% trypsin solution for 30 min at 37°C, followed by trituration with pipette in plating medium (DMEM with 10% fetal bovine serum, 10% F-12, and 1% penicillin–streptomycin). The dissociated neurons were plated onto poly-D-lysine-coated coverslips at a density of $1 \times 10^5$ cells per 35 mm dish. Following a 4-hr incubation period, the plating medium was replaced with Neurobasal medium (Thermo Fisher, 21103049) supplemented with 2% B-27 (Thermo Fisher, 17504044), 1% GlutaMax (Thermo Fisher, 35050061), and 1% penicillin–streptomycin (Invitrogen). Cytarabine (Sigma, C1768-100MG) was added after 2 days to a final concentration of 1 μM, to inhibit the proliferation of dividing non-neuronal cells. Half volume of the culture medium was replaced every 3–4 days with freshly prepared medium. The neurons were harvested at 14 days in vitro (DIV 14) for immunofluorescence experiments.

## cLTP and immunofluorescence

For cLTP, neurons were incubated with 300 μM glycine in a cell incubator set at 37°C with 5% $CO_2$ and 95% humidity for 3 min, followed by an additional 60 min without glycine at 37°C (*Wu et al., 2017*). Neurons were then fixed with 4% paraformaldehyde for 30 min on ice to preserve cell membranes. After washing with PBS, the neurons were penetrated with 0.3% Triton X-100 and blocked with 1% BSA in PBS for 30 min at RT. After that, neurons were incubated with primary antibodies (chicken anti-MAP2 antibody, 1:2000, Abcam, ab5392; mouse monoclonal anti-PSD95 antibody, 1:800, Abcam, ab192757; and rabbit monoclonal anti-NAT10 antibody, 1:800, Abcam, ab194297) overnight at 4°C. Neurons were then washed by PBS three times before incubation with fluorescent secondary antibodies for 2 hr at 37°C. The following secondary antibodies were employed: Chicken IgY (H+L) Secondary Antibody (A-11039, 1:500, Thermo Fisher), Rabbit IgG (H+L) Cross-Adsorbed Secondary Antibody (A-21244, 1:500, Thermo Fisher), Rabbit IgG (H+L) Highly Cross-Adsorbed Secondary Antibody (A32732, 1:500, Thermo Fisher), and Mouse IgG (H+L) Highly Cross-Adsorbed Secondary Antibody (A32727, 1:500, Thermo Fisher). After washing with PBS, the neurons were placed on mounting solution with DAPI (Beyotime, P0131). Confocal stacks were acquired from 3 to 6 individual fields on multiple coverslips per culture using a Leica SP8 confocal microscope. Two independent cultures were analyzed for each condition, including vehicle-treated control and glycine-treated neurons.

## Regular acRIP-seq for HOM

The acRIP-seq was performed following the previous studies (*Arango et al., 2018*; *Thalalla Gamage et al., 2021*). The total RNA was extracted from the HOM of mouse hippocampus using TRIzol, the quality of the RNA was evaluated through gel electrophoresis in order to ascertain its integrity and purity, and the concentration and purity of the RNA were detected. Total RNA samples, initially 100 μg, were treated with the rRNA depletion kit (New England Biolabs, Cat. No. E6310) to remove rRNA. After rRNA depletion, the RNA was subjected to immunoprecipitation using the anti-ac4C antibodies and the ac4C RIP kit (GenSeq, Cat No. GS-ET-005) according to the manufacturer's instructions. Briefly, total RNAs were randomly fragmented to ~200 nt by RNA Fragmentation Reagents (NEB, Cat No. M0348S). Protein A/G Dynabeads (Share-Bio, SB-PR001) were coupled to the anti-ac4C antibody by rotating at RT for 1 hr. The RNA fragments were then incubated with the beads-linked anti-ac4C antibodies (2 μg) in the 100 μl acRIP buffer (PBS containing 1% Triton X-100, 2 mM BSA, and 1 mM RNase inhibitor) and rotated at 4°C for 4 hr. After incubation, the RNA/antibody complexes were washed four times with the acRIP buffer, and then RNAs were eluted by elution buffer (containing 0.1 M glycine, 0.5% Triton X-100, pH 2.5–3.1) at RT for 10 min. Finally, add NaOH solution to adjust the pH of the elution product to neutral. Both the immunoprecipitated RNA and 1% input RNA are reverse transcribed with random hexamers, followed by RNA library preparation with Ultra II Directional RNA Library Prep Kit (New England Biolabs, Cat. No. E7760) according to the manufacturer's instructions.

The quality of RNA libraries was assessed using Agilent 2100 bioanalyzer and then subjected to the next-generation sequencing through the NovaSeq 6000 platform (Illumina). The quality of the sequencing data is measured using Q30 scores, and low-quality reads and 3' adaptors are removed using Cutadapt (v1.9.3). High-quality reads are then aligned to the reference genome using Hisat2 (v2.0.4). Raw counts are generated using HTSeq (v0.9.1) and normalized using edgeR (v3.16.5). Gene expression was quantified by logged transcript per million mapped reads ($\log_2$TPM) (with a cutoff

TPM >1). The ac4C peaks and mRNAs that were identified from at least two of the three biological replicates of HOM were considered reliable.

## Low-input acRIP-seq for SYN

The low-input acRIP-seq included some modifications, compared with the regular acRIP-seq. First, the RNA amount is relatively smaller (~2 µg, purified from the hippocampus of 10 mice) for low-input acRIP-seq, compared with regular acRIP-seq (~100 µg, purified from the hippocampus of 2 mice). Second, the total RNA purified from the SYN was not treated by the rRNA depletion kit before acRIP to avoid the non-specific RNA loss during the rRNA depletion process. Third, the RNA libraries for low-input acRIP-seq were generated by a different kit - SMARTer Stranded Total RNA-Seq Kit v2-Pico Input Mammalian (Takara, Cat. No. 634413). To increase the data reliability for low-input acRIP-seq, transcripts that were identified in our own sequencing and at least two of the three synaptosome databases (*Cajigas et al., 2012*; *Hafner et al., 2019*; *Niu et al., 2023*) were considered reliable and included in the analysis. Considering the principle of 3Rs (replacement, reduction, and refinement) for animal ethics, we used one biological replicate for each time point from control and memory mice, but each replicate consists of SYN RNA purified from 10 mice. The mitochondrially encoded mRNAs and long noncoding RNA (lncRNA) were filtered out before analysis of the acRIP-seq data since the current study focused on ac4C of regular mRNA in the SYN.

## Identification of ac4C peaks

Following RNA integrity assessments, acRIP enrichment verification, and library quality checks, raw sequencing reads from both acRIP and input samples underwent rigorous quality control. Initially, sequencing quality was evaluated using Q30 scores and FastQC to ensure base-level accuracy. Low-quality reads and 3′ adaptor sequences were identified and removed using Cutadapt software (v1.9.3), resulting in high-quality clean reads for further analysis.

Post-alignment filtering was applied to exclude reads that mapped to mitochondrial DNA (chrM) and non-concordant mate pairs, which often arise from sequencing artifacts or alignment errors. The remaining clean reads were then aligned to the mouse genome (mm10) using Hisat2 software (v2.0.4), with parameters allowing a maximum of five mismatches per read to ensure alignment accuracy above 95%. Ensembl gene annotations (Release 75) were incorporated to guide accurate mapping to annotated regions. To account for transcript variability, canonical transcript sequences from the UCSC genome browser were used to generate a Bowtie2 index, enabling transcriptome mapping. Reads were aligned in local mode, which is well-suited for fragmented RNA sequences.

Acetylation peaks (ac4C peaks) were identified using MACS software (v1.4.2) (*Zhang et al., 2008*), optimized for transcriptomic data. The shifting model and local lambda options were disabled to accommodate the unique characteristics of transcriptome data, and the genome size was set according to the total transcript base count. Input samples were used as controls to distinguish true ac4C-enriched peaks from background noise. The FE of ac4C peak was calculated through the MACS software by normalizing the read counts of acRIP ac4C peak to that of input ac4C peak. We established strict screening criteria in the peak calling process: adjusted $p < 0.05$ and FE >2. The adjusted p-value was acquired by MACS software using a Poisson distribution model after post hoc Benjamini–Hochberg analysis. The ac4C peaks were generated on the UCSC genome browser (https://genome.ucsc.edu/cgi-bin/hgGateway) using the IGV software (http://www.igv.org/).

## Sequence motif analysis

Sequence motifs were identified using HOMER (*Heinz et al., 2010*) and visualized using ggseqlogo package from R software. HOMER identifies motifs that are specifically enriched on memory set of sequences relative to control (e.g., peaks upstream and downstream sequences relative to the entire genome sequence) based primarily on the principle of ZOOPS scoring with hypergeometric enrichment.

## GO and functional category analysis

GO analysis to detect enrichment for GO terms in differential acetylation genes in R using the clusterProfiler (*Wu et al., 2021*), and the possible roles of these differential acetylation genes were annotated according to MF, BP, and CC. GO terms with a false-discovery rate adjusted p.adj of <0.05 were

visualized using R scripts to plot. The R package clusterProfiler to visualize the genes composing KEGG pathways, and p.adj <0.05 was used as the threshold of significant enrichment.

## Analysis of ac4C FE for individual mRNA

The FE of ac4C mRNA was reflected by the summary FE values of all ac4C peaks identified in the entire transcript including the 5′UTR, CDS, and 3′UTR. Averaging the cumulative distribution of these peaks offers a more comprehensive overview of the abundance of ac4C modifications across the mRNA molecule, avoiding the artificial increase of FE of long mRNAs (*Guca et al., 2024*). Additionally, in instances where ac4C modifications on mRNA are more dispersed or present in low abundance, a single prominent peak may not be observable; instead, multiple low-abundance peaks might emerge. In such scenarios, the accumulation of multiple peaks can enhance detection sensitivity, better capture dynamic changes, and mitigate biases inherent in individual experiments.

$$FE\,of\,ac4C = \Sigma peak\,(1+2+3\ldots\ldots+n)$$

## Analysis of ac4C FC for individual mRNA

The FC of ac4C mRNA is representative of the degree of change in the overall acetylation level of mRNA in the memory group (M) compared with the control group (C). To prevent a value of 0 in the denominator in the FC calculation, we added 1 to the FE of ac4C. The FC of ac4C mRNA was calculated using the following quantification index:

$$FC\,of\,ac4C = \frac{\left(FE\,of\,ac4C + 1\right)\,in\,M}{\left(FEof\,ac4C + 1\right)\,in\,C}$$

## Analysis of PPI network

The STRING (https://string-db.org/) (version 12.0) knowledge base was used to create a PPI network. The PPI network was generated with a confidence score of 0.7 and visualized using Cytoscape (version 3.10.1). Molecular Complex Detection (MCODE) (v2.0.3) algorithm was employed to identify the components of the densely connected network.

The following software and algorithms were used for the quantification and statistical analysis of ac4C epitranscriptome.

| Software and algorithms, application, authors | | |
|---|---|---|
| R v4.3.2 | Statistical computing and graphics. | *R Development Core Team, 2023* |
| Python v3.11 | Object-oriented programming language. | *Van Rossum and Drake, 2009* |
| ggplot2 | Creating graphics, based on The Grammar of Graphics. | *Wickham, 2016* |
| ggseqlogo | Generating publication-ready sequence logos using ggplot2. | *Wagih, 2017* |
| tidyverse | Collection of R packages designed for data science. | *Wickham et al., 2019* |
| ComplexHeatmap | Arrange multiple heatmaps and supports annotation graphics. | *Gu et al., 2016* |
| HOMER v4.11 | Motif discovery and gene sequencing analysis. | *Heinz et al., 2010* |
| bedtools | Compare large sets of genomic features. | *Quinlan and Hall, 2010* |
| cutadapt v1.9.3 | Remove sequencing data adaptor. | *Martin, 2011* |
| Hisat2 v2.04 | Mapping sequencing reads to a single reference genome. | *Zhang et al., 2021d* |
| MACS v1.4.2 | Identifying transcript factor-binding sites. | *Zhang et al., 2008* |
| Pandas v2.1.4 | Cleaning data. | *The pandas development team, 2026* |
| IGV | Interactive genome visualization application. | *Robinson et al., 2023* |

*Continued on next page*

*Continued*

| Software and algorithms, application, authors | | |
| --- | --- | --- |
| HTSeq (v0.9.1) | Analyzing high-throughput sequencing data. | *Anders et al., 2015* |
| edgeR (v3.16.5) | Empirical analysis of digital gene expression data. | *McCarthy et al., 2012* |
| CIBERSORT | Cell-type identification by estimating relative subsets of RNA transcripts. | *Newman et al., 2015* |

## Identification of relative abundance of gene expression in different cell types

To determine the cellular origin of the ac4C mRNAs, we performed a bioinformatic deconvolution analysis using the CIBERSORT algorithm (*Newman et al., 2015*). CIBERSORT is a computational method that uses a set of reference gene expression signatures to estimate the relative proportions of different cell types within a bulk RNA-sequencing dataset. This analysis is based on a machine learning approach, specifically $\nu$-support vector regression ($\nu$-SVR), which is robust against noise and confounding factors present in complex tissue samples. The analysis first required the construction of a high-quality reference signature matrix containing the gene expression profiles of pure cell populations. We constructed this matrix using a publicly available single-cell RNA-sequencing dataset of the adult mouse hippocampus from the McCarroll Lab (http://dropviz.org/) (*Saunders et al., 2018*). From this dataset, we extracted the expression signatures for four major hippocampal cell types: excitatory neurons, inhibitory neurons, astrocytes, and microglia, creating a signature matrix that reflects the unique transcriptomic profiles of these cells. Three groups of ac4C mRNAs (520 NAT10-independent MISA mRNAs, 717 NAT10-dependent MISA mRNAs, and 2629 NASA mRNAs that do not belong to MISA) were served as the input 'mixture' file for this analysis. We then ran the CIBERSORT algorithm in R, using our custom reference signature matrix to deconvolve the NASA mRNA list. The algorithm was run with 100 permutations to ensure statistical significance of the deconvolution results. The output of CIBERSORT provides an estimation of the relative contribution of each reference cell type to the total expression of the input gene set, which allowed us to infer which cell types were the primary contributors to the three groups of ac4C mRNAs.

## RT-qPCR

The cDNA was obtained through reverse transcription using the HiScript III RT SuperMix for qPCR kit (Vazyme, R323-01), followed by amplification using the ChamQ Universal SYBR qPCR Master Mix kit (Vazyme, Q711-02). Gene expression levels measured with RT-PCR were normalized to *Gapdh*. The primers used in qPCR for this study were as follows: *Gapdh*-forward, 5'gggtgtgaaccacgagaaat3'; *Gapdh*-reverse, 5'actgtggtcatgagcccttc3'; *Arc*-forward, 5'ggagggaggtcttctaccgtc3'; *Arc*-reverse, 5'cc cccacacctacagagaca3'; *Grin2b*-forward, 5'ctggtgaccaatggcaagcatg3'; *Grin2b*-reverse, 5'ggcacagaga agtcaaccacct3'; *Camk2a*-forward, 5'agccatcctcaccactatgctg3'; *Camk2a*-reverse, 5'gtgtcttcgtcctcaatg gtgg3'.

## Stereotaxic injection of AAV into mouse hippocampus

AAV2/9-Syn1-EGFP and AAV2/9-Syn1-EGFP-P2A-Cre were generated by Hanbio Technology (Shanghai) Corp, Ltd, with a titer of $10^{12}$/µl. We injected 0.3 µl AAV into each side of the mouse hippocampus. Six-week-old *Nat10*^flox/flox mice received an intraperitoneal injection of alphaxalone at a dosage of 40 mg/kg for anesthesia and then were immobilized in a stereotaxic frame provided by RWD Life Science. Similarly, AAV2/9-CAG-DIO-EGFP, generated by Hanbio Technology (Shanghai) Corp, Ltd, with a titer of $10^{12}$/µl, was injected into the hippocampus of 7-week-old *Nat10*^Cre/+ mice. The AAVs were delivered into mouse hippocampus through a stereotaxic injector at a flow rate of 0.1 µl/min. The precise coordinates for the stereotaxic injection were set to anteroposterior (AP) −2.3 mm, dorsoventral (DV) −1.8 mm, and mediolateral (ML) ±1.75 mm, relative to the Bregma. To induce Cre enzyme expression, mice were fed a custom-made tamoxifen-infused diet over a period of 1 week (*Tan et al., 2012*).

## Electrophysiological recording

Hippocampal slices were prepared as previously described (*Yin et al., 2013*). Briefly, mice (10 weeks old, male) were decapitated and transverse hippocampal slices (400 μm) were prepared using a Vibroslice (VT 1000S; Leica) in ice-cold ACSF. For field potential recording, the same ACSF was used for slice preparation and recording: 120 mM NaCl, 2.5 mM KCl, 1.2 mM $NaH_2PO_4$, 2.0 mM $CaCl_2$, 2.0 mM $MgSO_4$, 26 mM $NaHCO_3$ and 10 mM glucose. After sectioning, hippocampal slices were allowed to recover in the chamber for 30 min at 34°C and then for a further 2–8 hr at RT (25 ± 1°C). All solutions were saturated with 95% $O_2$/5% $CO_2$ (vol/vol).

The slices were placed in the recording chamber, which was superfused (3 ml/min) with ACSF at 32–34°C. fEPSPs were evoked in the CA1 stratum radiatum by stimulating the SC with a bipolar stimulating electrode (FHC) and recorded in current clamp using a HEKA EPC 10 amplifier (HEKA Elektronik) with ACSF-filled glass pipettes (1–3 MΩ). Test stimuli consisted of monophasic 100-μs pulses of constant current (intensity adjusted to produce 25% of the maximum response) at a frequency of 0.033 Hz. The strength of synaptic transmission was determined by measuring the initial amplitude of fEPSPs. L-LTP was induced by 4 trains of 100 Hz stimuli of 50 pulses each at 100 Hz, 10 s apart, at the same intensity as the test stimulus. The level of E-LTP / L-LTP was determined at an average of 55–60/115–120 min after HFS stimulation. To study glutamatergic transmission at SC-CA1 synapses, we recorded fEPSP input/output (I/O) curves at SC-CA1 synapses. To construct an I/O curve, fEPSPs were evoked by a series of incremental intensities (0.02–0.2 mA in 0.02 mA increments, 100 μs duration, at 0.033 Hz). Whole-cell patch-clamp recordings from CA1 neurons were visualized with infrared optics using an upright microscope equipped with a 40×water-immersion objective (BX51WI; Olympus) and an infrared-sensitive CCD camera. To analyze the intrinsic excitability of CA1 pyramidal neurons, the current clamp was used and neurons were held at 0 pA and then injected with step currents (duration: 500ms; increments: 20 pA; range: –100 to 400 pA; interval: 10 s). The input-output relationship was determined by plotting the injected currents against the number of action potentials. All data were acquired at a sampling rate of 10 kHz using PATCHMASTER version 2x90.1 software (HEKA Elektronik) and filtered off-line at 1 kHz.

## Generation and use of the ac4C website

The ac4Catlas website was constructed using HTML (a scripting language for front-end development), PHP (a scripting language for backend development), and MySQL (a structured query language for data operation). All the website codes were deployed in a Linux server equipped with CentOS environment. The raw data, processed data, and annotated files were stored in the 'Data Downloads' section. JBrowse (https://jbrowse.org) was integrated to visualize the ac4C epitranscriptomic data on genomic regions of interest. Briefly, ac4Catlas provides user-friendly interactive interfaces that allow researchers to freely browse and perform extended operations, such as searching, comparing, and downloading. Besides, the browse module stores detailed information of samples and corresponding datasets as well as ac4C levels of mRNA and differentially acetylated regions between groups.

## Statistical analysis

The cumulative distribution curve was analyzed by Kolmogorov–Smirnov test in R. The statistical analysis of percentage of ac4C summits occurring within CDS or UTRs was performed by chi-square test in R. The regular quantification data were shown as mean ± SEM unless otherwise mentioned. The sample size justification was based on the previous studies and met the requirement of statistics power. For normally distributed data with 1 experimental variable, statistical analyses were performed by parametric analysis: unpaired (two-tailed) Student's *t*-test for two groups and one-way ANOVA with the Tukey or Sidak multiple-comparison test for three groups. Data with >1 experimental variable were evaluated by two-way ANOVA, with post hoc Tukey, Dunnett, or Sidak multiple-comparison tests. All these statistical analyses were performed by the software of GraphPad Prism 10. Statistically significant difference was indicated as follows: \*\*\*$p < 0.001$, \*\*$p < 0.01$, and \*$p < 0.05$.

## Code availability

Custom code used in this paper is available at https://github.com/YinDMlab/ac4CRIP-seq-analysis copy archived at *YinDMlab, 2024*.

## Acknowledgements

We thank CloudSeq Inc (Shanghai, China) for providing us with technical support on acRIP-seq. This work was supported by STI2030-Major Projects (Grant No. 2021ZD0202500), National Natural Science Foundation of China (Grant No.32500917), Natural Science Foundation of Chongqing, China (Grant No. 2022NSCQ-MSX5813), ECNU Multifunctional Platform for Innovation (001), and the Proton Cluster Incubation Project of the National Natural Science Foundation of China (Grant No. 2025GZRFY01).

## Additional information

### Funding

| Funder | Grant reference number | Author |
| --- | --- | --- |
| STI2030-Major Projects | Grant No. 2021ZD0202500 | Hai-Qian Zhou |
| National Natural Science Foundation of China | Grant No. 32500917 | Hai-Qian Zhou |
| Proton Cluster Incubation Project of the National Natural Science Foundation of China | Grant No. 2025GZRFY01 | Hai-Qian Zhou |
| Natural Science Foundation of Chongqing, China | Grant No. 2022NSCQ-MSX5813 | Wei-Peng Lin |
| ECNU Multifunctional Platform for Innovation | 001 | Dong-Min Yin |

The funders had no role in study design, data collection, and interpretation, or the decision to submit the work for publication.

### Author contributions

Hai-Qian Zhou, Conceptualization, Data curation, Software, Formal analysis, Funding acquisition, Validation, Visualization, Methodology, Writing – original draft, Project administration, Writing – review and editing; Zhen Zhu, Jia-Wei Zhang, Data curation, Software, Writing – review and editing; Wei-Peng Lin, Data curation, Writing – review and editing; Hao-JY Jin, Yang-Yang Ding, Data curation; Shuai Liu, Writing – review and editing; Dong-Min Yin, Conceptualization, Supervision, Funding acquisition, Writing – original draft, Writing – review and editing

### Author ORCIDs

Hai-Qian Zhou ⬚ https://orcid.org/0000-0003-1439-3506
Hao-JY Jin ⬚ https://orcid.org/0009-0008-4897-4791
Shuai Liu ⬚ https://orcid.org/0000-0001-5849-489X
Dong-Min Yin ⬚ https://orcid.org/0000-0001-9329-050X

### Ethics

All animal procedures were approved by the Institutional Animal Care and Use Committee of East China Normal University (Approval No. m20220604) and were conducted in accordance with institutional guidelines for the care and use of laboratory animals. All efforts were made to minimize animal suffering.

Reviewer #1 (Public review): https://doi.org/10.7554/eLife.108995.3.sa1
Reviewer #2 (Public review): https://doi.org/10.7554/eLife.108995.3.sa2
Author response https://doi.org/10.7554/eLife.108995.3.sa3

# Additional files

## Supplementary files
Supplementary file 1. acRIP-seq sequencing library information and lists of ac4C-modified mRNAs in mouse hippocampus.

MDAR checklist

## Data availability
All data are available in the main text, supplementary or source data files. The original data of acRIP-seq is accessible via the website (http://ac4Catlas.com/) or GEO database (GSE261922). All unique materials used are readily available from the corresponding author upon reasonable requests.

The following dataset was generated:

| Author(s) | Year | Dataset title | Dataset URL | Database and Identifier |
|---|---|---|---|---|
| Zhou H, Zhang J, Zhu Z, Chen D, Yin D | 2024 | Dynamic change of ac4C in mouse hippocampus during learning and memory | https://www.ncbi.nlm.nih.gov/geo/query/acc.cgi?acc=GSE261922 | NCBI Gene Expression Omnibus, GSE261922 |

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
