## [Editor Report · eLife Assessment]

Recent studies have shown that mRNA can be acetylated (ac4c), altering mRNA stability and translation efficiency; however, the role of mRNA acetylation in the brain remains unexplored. In this **important** study, the authors demonstrate that ac4c occurs in synaptically localised mRNAs, mediated by NAT10. Conditional reduction of NAT10 protein levels led to decreases in ac4c of mRNAs and deficits in synaptic plasticity and memory. These **solid** results suggest that mRNA acetylation may play a role in memory consolidation.

---

## [Referee Report · Reviewer #1 (Public review)]

Summary:

RNA modification has emerged as an important modulator of protein synthesis. Recent studies found that mRNA can be acetylated (ac4c), which can alter mRNA stability and translation efficiency. The role of ac4c mRNA in the brain has not been studied. In this paper, the authors convincingly show that ac4c occurs selectively on mRNAs localized at synapses, but not cell wide. The ac4c "writer" NAT10 is highly expressed in hippocampal excitatory neurons. Using NAT10 conditional KO mice, decreasing levels of NAT10 resulted in decreases in ac4c of mRNAs and also showed deficits in LTP and spatial memory. These results reveal a potential role for ac4c mRNA in memory consolidation.

This is a new type of mRNA regulation that seems to act specifically at synapses, which may help elucidate the mechanisms of local protein synthesis in memory consolidation. Overall, the studies are well carried out and presented. The precise mRNAs that require ac4c to carry out memory consolidation is not clear, but is an important focus of future work. The specificity of changes occurring only at the end of training, rather than after each day of training is interesting and also warrants further investigation. This timeframe is puzzling because the authors show that ac4c can dynamically increase within 1hr after cLTP.

Strengths:

(1) The studies show that mRNA acetylation (ac4c) occurs selectively at mRNAs localized to synaptic compartments (using synaptoneurosome preps).

(2) The authors identify a few key mRNAs acetylated involved in plasticity and memory - eg Arc.

(3) The authors show that Ac4c is induced by learning and neuronal activity (cLTP).

(4) The studies show that the ac4c "writer" NAT10 is expressed in hippocampal excitatory neurons and may relocated to synapses after cLTP/learning induction.

(5) The authors used floxed NAT10 mice injected with AAV-Cre in the hippocampus (NAT10 cKO) to show that NAT10 may play a role in LTP maintenance and memory consolidation (using the Morris Water Maze).

Weaknesses:

(1) The NAT10 cKO mice are useful to test the causal role of NAT10 in ac4a and plasticity/memory but all the experiments used AAV-CRE injections in the dorsal hippocampus that showed somewhat modest decreases in total NAT10 protein levels. For these experiments, it would be better to cross the NAT10 floxed animals to CRE lines where better knock down of NAT10 can be achieved postnatally in specific neurons, with less variability.

(2) Because knock down is only modest (~50%), it is not clear if the remaining ac4c on mRNAs is due to remaining NAT10 protein or due to alternative writer (as the authors pose).

---

## [Referee Report · Reviewer #2 (Public review)]

This is an interesting study that shows that mRNA acetylation at synapses is dynamically regulated at synapses by spatial memory in the mouse hippocampus. The dynamic changes of ac4C-mRNAs regulated by memory were validated by methods including ac4C dot-blot and liquid 13 chromatography-tandem mass spectrometry (LC-MS/MS).

---

## [Author Response]

The following is the authors’ response to the original reviews.

**Public Reviews:**

**Reviewer #1 (Public review):**
(1) The authors use a confusing timeline for their behavioral experiments, i.e., day 1 is the first day of training in the MWM, and day 6 is the probe trial, but in reality, day 6 is the first day after the last training day. So this is really day 1 post-training, and day 20 is 14 days post-training.

We have revised the timeline accordingly. Briefly, mice were trained in the Morris water maze (MWM) with a hidden platform for five consecutive days (training days 1–5). Probe tests were then conducted on day 6 and day 20, which correspond to post-training day 1 and post-training day 15, respectively. We clearly stated as such in the revised manuscript (see results, line 108 – 113) and figure S1 (see figure legend, line 1747 – 1749).

(2) The authors inaccurately use memory as a term. During the training period in the MWM, the animals are learning, while memory is only probed on day 6 (after learning). Thus, day 6 reflects memory consolidation processes after learning has taken place.

We have revised the manuscript to distinguish between "learning" and "memory". We refer to the performance during the 5-day training period as "spatial learning" and restrict the term "memory" to the probe tests on day 6, which reflect memory consolidation after learning has taken place.

(3) The NAT10 cKO mice are useful... but all the experiments used AAV-CRE injections in the dorsal hippocampus that showed somewhat modest decreases... For these experiments, it would be better to cross the NAT10 floxed animals to CRE lines where a better knockdown of NAT10 can be achieved, with less variability.

We want to clarify the reason for using AAV-Cre injection rather than Cre lines. Indeed, we attempted to generate Nat10 conditional knockouts by crossing Nat10^flox/flox^ mice with several CNS-specific Cre lines. Crossing with Nestin-Cre and Emx1-Cre resulted in embryonic and premature lethality, respectively, consistent with the essential housekeeping function of NAT10 during neurodevelopment. We will use the Camk2α-Cre line which starts to express Cre after postnatal 3 weeks specifically in hippocampal pyramidal neurons (Tsien et al., 1996).

(4) Because knockdown is only modest (~50%), it is not clear if the remaining ac4c on mRNAs is due to remaining NAT10 protein or due to an alternative writer (as the authors pose).

Our results suggest the existence of alternative writers. As shown in Figure 6D, we identified a population of "NAT10-independent" MISA mRNAs (present in MISA but not downregulated in NASA). Remarkably, these mRNAs possess a consensus motif (RGGGCACTAACY) that is fundamentally different from the canonical NAT10 motif (AGCAGCTG). This distinct motif usage suggests that the residual ac4C signals are not merely due to incomplete knockdown of NAT10, but reflect the activity of other, as-yet-unidentified ac4C writers. We will perform ac4C immunostaining in Nat10-reporter mice which express red fluorescent proteins in Nat10-positive cells. The results that ac4C is expressed in both Nat10-positive and negative cells will support the presence of as-yet-unidentified ac4C writers.

**Reviewer #2 (Public review):**
(1) It is known that synaptosomes are contaminated with glial tissue... So the candidate mRNAs identified by acRIP-seq might also be mixed with glial mRNAs. Are the GO BP terms shown in Figure 3A specifically chosen, or unbiasedly listed for all top ones?

This reviewer is correct that some ac4C-mRNAs identified by acRIP-seq from the synaptosomes are highly expressed in astrocytes, such as Aldh1l1, ApoE, Sox9 and Aqp4 (see list of ac4C-mRNAs in the synaptosomes, Table S3). In agreement, we found that NAT10 was also expressed in astrocyte in addition to neurons. We have provided a representative image showing NAT10-Cre expression in astrocytes in the revised manuscript (Figure 4F and H). In the figure 3A of original submission, we showed 10 out of 16 top BP items for MISA mRNAs. In the figure 3A of revised manuscript, we showed all the top 16 BP items for MISA mRNAs, which are unbiasedly chosen (also see Table S4).

(2) Where does NAT10-mediated mRNA acetylation take place within cells generally? Is there evidence that NAT10 can catalyze mRNA acetylation in the cytoplasm?

The previous studies from non-neuronal cells showed that NAT10 can catalyze mRNA acetylation in the cytoplasm and enhance translational efficiency (Arango et al., 2018; Arango et al., 2022). In this study, we showed that mRNA acetylation occurred both in the homogenates and synapses (see ac4C-mRNA lists in Table S2 and S3). However, spatial memory upregulated mRNA acetylation mainly in the synapses rather than in the homogenates (Fig. 2 and Fig. S2).

(3) "The NAT10 proteins were significantly reduced in the cytoplasm (S2 fraction) but increased in the PSD fraction..." The small increase in synaptic NAT10 might not be enough to cause a decrease in soma NAT10 protein level.

We showed that the NAT10 protein levels were increased by one-fold in the PSD fraction, but were reduced by about 50% in the cytoplasm after memory formation (Fig. 5J and K). The protein levels of NAT10 in the homogenates and nucleus were not altered after memory formation (Fig. 5F and I). Due to these facts, we hypothesized that NAT10 proteins may have a relocation from cytoplasm to synapses after memory formation, which was also supported by the immunofluorescent results from cultured neurons (Fig. S4). However, we agree with this reviewer that drawing such a conclusion may require the time-lapse imaging of NAT10 protein trafficking in living animals, which is technically challenging at this moment.

(4) It is difficult to separate the effect on mRNA acetylation and protein mRNA acetylation when doing the loss of function of NAT10.

This is a good point. We agree with this reviewer that NAT10 may acetylate both mRNA and proteins. We examined the acetylation levels of a-tubulin and histone H3, two substrate proteins of NAT10 in the hippocampus of Nat10 cKO mice. As shown in Fig S5C, E, and F, the acetylation levels of a-tubulin and histone H3 remained unchanged in the Nat10 cKO mice, likely due to the compensation by other protein acetyltransferases. In contrast, mRNA ac4C levels were significantly decreased in the Nat10 cKO mice (Figure S5G–H). These results suggest that the memory deficits seen in Nat10 cKO mice may be largely due to the impaired mRNA acetylation. Nonetheless, we believe that developing a new technology which enables selective erasure of mRNA acetylation would be helpful to address the function of mRNA acetylation. We discussed these points in the MS (see discussion, line 582-589).

Reference

Arango, D., Sturgill, D., Alhusaini, N., Dillman, A. A., Sweet, T. J., Hanson, G., Hosogane, M., Sinclair, W. R., Nanan, K. K., & Mandler, M. D. (2018). Acetylation of cytidine in mRNA promotes translation efficiency. Cell, 175(7), 1872-1886. e1824.

Arango, D., Sturgill, D., Yang, R., Kanai, T., Bauer, P., Roy, J., Wang, Z., Hosogane, M., Schiffers, S., & Oberdoerffer, S. (2022). Direct epitranscriptomic regulation of mammalian translation initiation through N4-acetylcytidine. Molecular cell, 82(15), 2797-2814. e2711.

Tsien, J. Z., Chen, D. F., Gerber, D., Tom, C., Mercer, E. H., Anderson, D. J., Mayford, M., Kandel, E. R., & Tonegawa, S. (1996). Subregion-and cell type–restricted gene knockout in mouse brain. Cell, 87(7), 1317-1326.